# MetaSlot: Break Through the Fixed Number of Slots in Object-Centric Learning

**Hongjia Liu[1]    Rongzhen Zhao[1]***    **Haohan Chen[2]    Joni Pajarinen[1]**

[1]Department of Electrical Engineering and Automation, Aalto University, Espoo, Finland
[2]Department of Computer Science, Sichuan University, Chengdu, China
`{hongjia.liu, rongzhen.zhao, joni.pajarinen}@aalto.fi`
`sajel@stu.scu.edu.cn`

## Abstract

Learning object-level, structured representations is widely regarded as a key to better generalization in vision and underpins the design of next-generation Pre-trained Vision Models (PVMs). Mainstream Object-Centric Learning (OCL) methods adopt Slot Attention or its variants to iteratively aggregate objects' super-pixels into a fixed set of query feature vectors, termed slots. However, their reliance on a static slot count leads to an object being represented as multiple parts when the number of objects varies. We introduce MetaSlot, a plug-and-play Slot Attention variant that adapts to variable object counts. MetaSlot (i) maintains a codebook that holds prototypes of objects in a dataset by vector-quantizing the resulting slot representations; (ii) removes duplicate slots from the traditionally aggregated slots by quantizing them with the codebook; and (iii) injects progressively weaker noise into the Slot Attention iterations to accelerate and stabilize the aggregation. MetaSlot is a general Slot Attention variant that can be seamlessly integrated into existing OCL architectures. Across multiple public datasets and tasks–including object discovery and recognition–models equipped with MetaSlot achieve significant performance gains and markedly interpretable slot representations, compared with existing Slot Attention variants. The code is available at https://github.com/lhj-lhj/MetaSlot.

## 1   Introduction

Human intelligence is rooted in limited perceptual experience–especially visual information–which enables it to demonstrate outstanding transfer and generalization abilities in entirely new task scenarios [1, 2]. In recent years, major breakthroughs in embodied intelligence have further underscored the inevitable trend of artificial intelligence moving into the physical world [3–5]. However, the key challenge in achieving high-level cognitive reasoning [6, 7] and compositional generalization [8, 9] lies in transforming visual inputs into structured, discrete, and independent object-level representations [10–12], thereby granting agents a deep understanding of physical objects and their dynamic relations [13, 14].

Object-Centric Learning (OCL) has rapidly developed against this backdrop. Its goal is to extract object-level structured representations in an unsupervised manner, rather than relying on attribute-level features or global scene features. Among numerous methods [15–20], Slot Attention (SA) [11] is currently the most influential and widely adopted. Through a competition mechanism among slots, it iteratively clusters distributed scene representations into several object-oriented feature vectors, named slots; each slot can then be decoded separately [11, 21], or all slots can be decoded jointly in an autoregressive fashion [22, 23] to produce semantically consistent segmentation masks. This

---

*Corresponding author.

39th Conference on Neural Information Processing Systems (NeurIPS 2025).

strategy not only efficiently captures object-level information but also lays a solid foundation for subsequent physical reasoning and relational modeling [24–26].

Nevertheless, classic Slot Attention [11] still suffers from two key limitations: (1) the number of slots must be preset as a fixed hyper-parameter; (2) slot initialization relies on random sampling. The former conflicts with the dynamic variability of object counts in real visual scenes, easily leading to under-segmentation or over-segmentation and harming the identifiability of the representations [27, 28]; the latter often results in object-centric representations that lack a clear correspondence with true object concepts [29]. Overall, a fixed slot count and random initialization are equivalent to imposing inappropriate prior assumptions on the latent space, making the model more prone to sub-optimal solutions and limiting its generalization capability.

Vector quantization (VQ) [30] offers a viable pathway: It has recently shown great value in generative modeling by enabling models to extract and reuse semantic structural patterns [31–33]. Inspired by this insight, we incorporate a VQ codebook that supplies globally shared object prototypes, guiding slot initialization structurally; meanwhile, we prune duplicate slots to provide explicit semantic cues about "objects" from the very start of the aggregation process. In particular, this idea of "object prototypes" echoes Plato's "world of forms" [34]: every concrete object in the perceptual world is a projection of some eternal and perfect ideal form. Analogously, we regard each prototype vector in the VQ codebook as an idealized object concept, whereas the input features are concrete mappings of these forms. Based on this intuition, we propose MetaSlot, a novel object-centric learning framework that employs a unified prior of global prototype slots and a dynamically adaptive two-stage aggregation method to flexibly match the slot count to the objects present in a scene. Specifically, our study introduces two important technical innovations:

**Dynamic slot allocation.** To address the above limitations, we design the MetaSlot framework to adaptively adjust the number of slots through two-stage aggregation. First, to match input features with the prototype codebook, we perform initial aggregation using Slot Attention in first stage. The resulting slot vectors are then matched to the global discrete codebook, producing semantically consistent discrete slot indices. Next, to allow the model to adjust the effective slot count according to scene complexity, we apply a de-duplication operation to slot indices that correspond to the same prototype, retaining only distinct prototype slots as the initialization for second stage. The object-aware initial slots are then fed into a mask slot attention module for a second aggregation stage, enabling fine-grained object-level assignment. Throughout this stage, the aggregator applies an attention mask to redundant slots and shares the same weights as the first-stage Slot Attention.

**Consistent prototypes and stable optimization.** To ensure that all parts of the same object converge to a consistent prototype, we use the final slot representations obtained from the second stage to update the codebook. Simultaneously, we employ a k-means-based exponential moving average (EMA) strategy to update the codebook stably and suppress high variance in the early training phase. In addition, we introduce a progressive noise-injection mechanism during training as implicit simulated annealing [35], further reinforcing efficient alignment between the prototype prior and the posterior over targets in latent space.

In short, MetaSlot leverages a global VQ codebook of object prototypes. Slot Attention clusters features, aligns them with their nearest prototypes, prunes duplicates, and passes the resulting semantically rich slots to a second aggregation stage. Moreover, injecting progressive noise during this stage helps stabilize convergence, yielding robust and accurate object representations. Our work makes three primary contributions: (i) MetaSlot module: We devise a two-stage aggregation framework that couples a global vector-quantized codebook with a slot-masking mechanism, enabling dynamic slot allocation for arbitrary numbers of objects. (ii) Progressive noise injection: By injecting gradually diminishing Gaussian noise throughout the Slot Attention iterations, MetaSlot both accelerates convergence and stabilizes the aggregation. (iii) Large-scale validation: Extensive experiments across diverse vision tasks and datasets show that MetaSlot yields substantial improvements on key metrics and exhibits strong adaptability to a wide range of scenes.

## 2 Method

In this section, we present **MetaSlot** with (i) First-stage Aggregation for Prototype-guided Pruning; (ii) Second-stage Aggregation with Mask-guided Refinement; and (iii) Prototype update via mini-

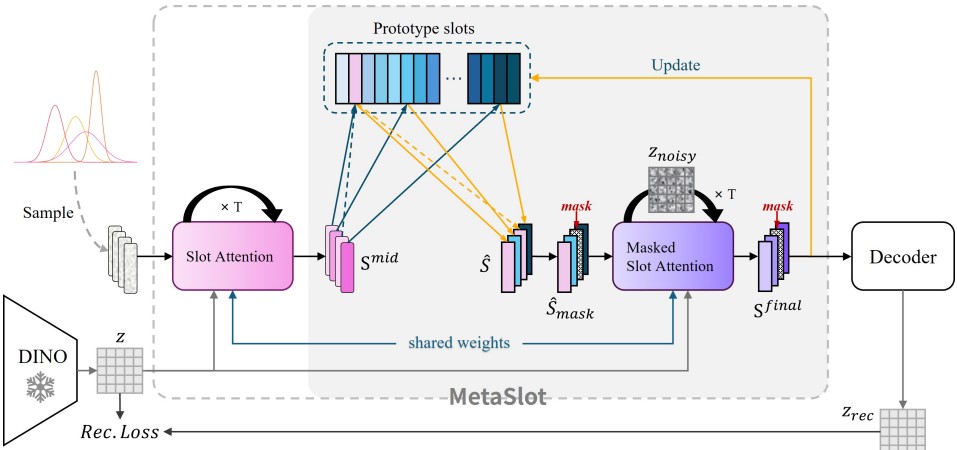

Figure 1: Overview of the MetaSlot framework (depicted on the DINOSAUR backbone for clarity; agnostic to the underlying object-centric architecture). (i) We build and continually update a codebook of "prototype slots" by vector-quantizing slots sampled across the dataset. (ii) Input features $\boldsymbol{Z}$ are first aggregated via Slot Attention to produce an intermediate slot set $\boldsymbol{S}^{\mathrm{mid}}$; we then remove duplicate slots in $\boldsymbol{S}^{\mathrm{mid}}$ by matching them against the prototype slots, yielding the masked subset $\hat{\boldsymbol{S}}_{\mathrm{mask}}$. (iii) Finally, $\hat{\boldsymbol{S}}_{\mathrm{mask}}$ is passed through Masked Slot Attention with progressively attenuated noise to generate the refined slots $\boldsymbol{S}^{\mathrm{final}}$, which are then decoded to reconstruct the original input.

batch K-means. Notably, the two aggregation modules share weights and are jointly trained. In addition, we include pseudocode in Appendix A to provide additional implementation details.

## 2.1 Background

**Slot Attention (SA).**   Slot Attention (SA) [11] transforms a set of input features $\boldsymbol{Z} \in \mathbb{R}^{N \times D}$ into $K$ object-centric representations $\boldsymbol{S} \in \mathbb{R}^{K \times D}$ through iterative cross-attention. Each iteration, slots compete by applying a softmax over themselves to claim parts of the visual input, and each slot's incremental information $\tilde{\boldsymbol{S}}$ is computed as the attention-weighted sum of visual feature vectors.

$$\tilde{\boldsymbol{S}} = f_{\phi_{\mathrm{attn}}}(\boldsymbol{S}, \boldsymbol{Z}) = \left(\frac{A_{i,j}}{\sum_{l=1}^{N} A_{l,j}}\right)^{\top} \cdot v(\boldsymbol{Z}), \quad \text{where } \boldsymbol{A} = \mathrm{softmax}\left(\frac{k(\boldsymbol{Z}) \cdot q(\boldsymbol{S})^{\top}}{\sqrt{D}}\right) \in \mathbb{R}^{N \times K}.$$
(1)

In the slot update stage, the incremental information $\tilde{\boldsymbol{S}}$ and the previous slot state $\boldsymbol{S}^{(t)}$ are fed into a Gated Recurrent Unit (GRU) [36]. The GRU output is then refined by a small MLP, yielding the updated slot state:

$$\boldsymbol{S}^{(t+1)} = \mathrm{MLP}\big(\mathrm{GRU}(\boldsymbol{S}^{(t)}, \tilde{\boldsymbol{S}})\big).$$
(2)

After $T$ iterations, the final slots $\boldsymbol{S}^{T}$ are employed as the object-centric representation passed to downstream modules. Crucially, these slots are randomly initialized from a learnable Gaussian distribution $\mathcal{N}(\boldsymbol{\mu}, \mathrm{diag}(\boldsymbol{\sigma}))$.

## 2.2 First-Stage Aggregation for Prototype-guided Pruning

In the first-stage aggregation, MetaSlot performs global prototype alignment, merging all features of an object into one slot, pruning duplicate slots, and yielding a compact, semantically coherent basis for later fine-grained aggregation.

To obtain prototype slot representations from the codebook that faithfully capture the input features, we first perform a preliminary aggregation of the feature maps, producing a set of softly assigned intermediate slots $\boldsymbol{S}^{\mathrm{mid}}$. Each intermediate slot is then matched to the global discrete codebook $\mathcal{E}$ via a nearest-neighbour search, yielding the semantically aligned discrete slots $\hat{\boldsymbol{S}}$.

Given the intermediate slots $\boldsymbol{S}^{\mathrm{mid}}$ produced by the original Slot Attention [11], the fixed number of slots can lead to a single object's features being scattered across multiple slots. To resolve this, prototype matching maps every slot encoding the same object onto a shared prototype, thereby eliminating redundancy. As a result, among slots with identical indices only the unique prototypes $\hat{\boldsymbol{S}}_{\mathrm{mask}}$ are retained, and this compact set is used to initialize the next stage.

**Intermediate slots.**   As in the original formulation, we sample the initial slots $\boldsymbol{S}^{(0)} \in \mathbb{R}^{N \times D}$ from a learnable Gaussian $\mathcal{N}(\boldsymbol{\mu}, \mathrm{diag}(\boldsymbol{\sigma}))$, and then perform $\boldsymbol{T}$ iterative slot updates on the input feature set $\boldsymbol{Z} \in \mathbb{R}^{F \times D}$.

$$\boldsymbol{S}^{\mathrm{mid}} = \mathrm{SlotAttn}(\boldsymbol{Z}, \boldsymbol{S}^{(0)}, \boldsymbol{T}). \tag{3}$$

**Nearest-neighbour quantisation.**   Let $\mathcal{E} = \{\boldsymbol{e}_k \in \mathbb{R}^D\}_{k=1}^{K}$ denote a global codebook of $K$ prototypes. For each intermediate slot vector $\boldsymbol{s}_i^{\mathrm{mid}}$, we find the index of its nearest prototype and replace it accordingly:

$$idx_i \;=\; \arg\min_k \left\| \boldsymbol{s}_i^{\mathrm{mid}} - \boldsymbol{e}_k \right\|_2, \qquad \hat{\boldsymbol{s}}_i \;=\; \boldsymbol{e}_{idx_i}, \tag{4}$$

where $\hat{\boldsymbol{s}}_i$ is the quantised slot vector, set to the prototype $\boldsymbol{e}_{idx_i}$.

**Duplicate-removal mask.**   Slots that pick the same prototype are treated as redundant. We mark the first occurrence and mask out the rest:

$$smask_i \;=\; \begin{cases} 1, & \text{if } idx_i \text{ is the first hit,} \\ 0, & \text{otherwise.} \end{cases} \tag{5}$$

The surviving slot set is $\hat{\boldsymbol{S}}_{\mathrm{mask}} = \{\hat{\boldsymbol{s}}_i \mid smask_i = 1\} \subseteq \mathbb{R}^{\hat{N} \times d}$ with $\hat{N} \leq N$.

### 2.3   Second-Stage Aggregation with Mask-guided Refinement

In the second-stage aggregation, we refine the pruned prototypes via masked attention with annealed noise injection, producing the final semantically coherent slot set $\boldsymbol{S}^{\mathrm{final}}$.

We re-initialize the slot states with the pruned prototype set $\hat{\boldsymbol{S}}_{\mathrm{mask}}$ and perform $T$ iterations of attention-based updates. At each step, the raw attention logits are masked by the binary slot mask $smask$, so that only retained prototypes participate in computing the attention-weighted slot increments. To alleviate the "cold-start" misalignment between prior prototypes and current inputs, we also inject progressively isotropic Gaussian noise into the features before each iteration, with variance $\alpha_t^2$ linearly annealed from $\sigma_{\mathrm{noise}}^2$ to zero. This implicit simulated annealing encourages early exploration and late-stage convergence, yielding the final semantically coherent slot set $\boldsymbol{S}^{\mathrm{final}}$.

**Implicit Simulated Annealing via Noise Injection**   In the classic Slot Attention module, slots aggregate visual features through iterative attention and GRU updates. However, when prototype slots are generated by offline vector quantization (VQ), initial misalignment often occurs between the prior slots and posterior slots derived from the current input in the latent space. To reduce this "cold-start" distance, we explicitly inject decreasing noise into the features before each iteration. This strategy can be interpreted as a form of implicit simulated annealing: injecting large-magnitude noise at early stages relaxes the entropy constraints of soft matching, encouraging exploration among slots; gradually reducing noise at later stages facilitates convergence to precise alignment. At any iteration step $t$, we add isotropic Gaussian noise to the features $\boldsymbol{Z} \in \mathbb{R}^{N \times D}$:

$$\boldsymbol{Z}_{\mathrm{noise}}^{(t)} \;=\; \boldsymbol{Z}^{(t)} + \boldsymbol{\xi}_{\mathrm{iso}}^{(t)}, \qquad \boldsymbol{\xi}_{\mathrm{iso}}^{(t)} \sim \mathcal{N}\big(\boldsymbol{0}, \alpha_t^2 \mathbf{I}_C\big), \tag{6}$$

where

$$\alpha_t \;=\; \sigma_{\mathrm{noise}}\Big(1 - \tfrac{t}{T-1}\Big). \tag{7}$$

Eq. (7) corresponds to a gradual decrease in temperature $\tau \propto \alpha_t^{-2}$, and $\sigma_{\text{noise}}$ is a tunable hyperparameter that specifies the initial standard deviation of the injected isotropic Gaussian noise (i.e. the noise amplitude at $t = 0$) before annealing.

**Masked Slot Attention (MSA).**   To ensure that only surviving slots steer the refinement, we introduce Masked Slot Attention (MSA). At each iteration, a binary mask $smask$ zeros out rows corresponding to duplicate slots, so the attention-weighted update is computed solely from the retained, semantically meaningful prototype slots. This prevents any duplicate-induced interference. It is worth noting that the MSA shares the same set of weights with the SA used in the first-stage aggregation. For each iteration $t = 0, \ldots, T - 1$ we compute the MSA:

$$\tilde{\boldsymbol{S}}^{(t)} = f_{\phi_{\text{attn}}}\big(\boldsymbol{S}^{(t)}, \boldsymbol{Z}_{\text{noise}}^{(t)}\big) = \Big(\frac{\tilde{A}_{i,j}^{(t)}}{\sum_{l=1}^{N} \tilde{A}_{l,j}^{(t)}}\Big)^{\top} \cdot v(\boldsymbol{Z}_{\text{noise}}^{(t)}), \qquad \tilde{A}_{i,j}^{(t)} = smask_i\, A_{i,j}^{(t)}, \qquad (8)$$

with

$$\boldsymbol{A}^{(t)} = \text{softmax}\Big(\frac{k(\boldsymbol{Z}_{\text{noise}}^{(t)}) \cdot q(\boldsymbol{S}^{(t)})^{\top}}{\sqrt{D}}\Big) \in \mathbb{R}^{N \times K}, \qquad (9)$$

where $q, k, v \in \mathbb{R}^{D}$ are the linearly projected queries, keys and values.

Finally, the slot states are updated as in Eq. (2), yielding $\boldsymbol{S}^{(t+1)}$.

**Gradient Truncation and Bi-level Optimization.**   Because the vector-quantization (VQ) mechanism truncates gradients–producing instability between Two-stage Slot Aggregation iterations, we stop the gradient flow at the first Slot-aggregation stage. In addition, inspired by the bi-level optimization strategy [29], we further detach gradients during the first $T - 1$ iterations of the second Slot-aggregation stage.

Let $\boldsymbol{S}_2^{(T)}$ be the slots after the $t$-th refinement step in second-stage aggregation. Thus all paths that reach the encoder features $\boldsymbol{Z}$ through $\boldsymbol{S}_1^{(0)}, \ldots, \boldsymbol{S}_2^{(T-1)}$ are detached, and only the T-th (final) refinement step $\boldsymbol{S}_2^{(T)}$ contributes gradients.

### 2.4   Prototype update via mini-batch *K*-means.

To encourage the codebook slots to converge toward identifiable slot prototypes, we use the final $\boldsymbol{S}_2^{(T)}$ to update the codebook. To ensure stable updates to the codebook, we adopt a K-means-based exponential moving average (EMA) update strategy. Specifically, at each training step we shift every prototype $\boldsymbol{e}_k$ toward the mini-batch centroid $\boldsymbol{c}_k$ computed over $\boldsymbol{S}_2^{(T)}$:

$$\boldsymbol{e}_k \;\leftarrow\; (1 - \eta)\,\boldsymbol{e}_k \;+\; \eta\,\boldsymbol{c}_k, \qquad (10)$$

where $\eta \in (0, 1]$ is a small learning rate.

Prototypes that remain unselected for a predefined *timeout* window are marked as dead. For each such code we sample a replacement vector $\tilde{\boldsymbol{c}}$ from the current mini-batch by choosing the slot that is least similar (cosine distance) to all active prototypes, and reset the dead code via

$$\boldsymbol{e}_k \;\leftarrow\; \tilde{\boldsymbol{c}}. \qquad (11)$$

## 3   Related Work

In recent years, unsupervised representation learning has made significant progress, with Slot Attention (SA) [11] playing a pivotal role in advancing this field. SA learns distinct latent representations for each object in an image through an iterative mechanism, and these latent "slots" can subsequently be decoded back into pixel space. Early slot-based methods [11, 21–23, 37] typically employed simple small-scale CNNs [38] or pre-trained ResNet models [39] as feature encoders, and used Spatial Broadcast Decoders [40] or Vision Transformers [41] as decoders, with tests mainly conducted on synthetic datasets. Recent approaches such as SlotDiffusion [42] and LSD [43] integrate SA with diffusion-model decoders [44, 45]. DINOSAUR and its variants [46–48] constructs reconstruction objectives based on DINO [49, 50] features to enhance object discovery in real-world data.

To enhance Slot Attention's intrinsic object-awareness, BO-QSA [29] introduces bi-level mechanisms and slot-level initialization, whereas ISA [51] incorporates pose-equivariance within its attention and

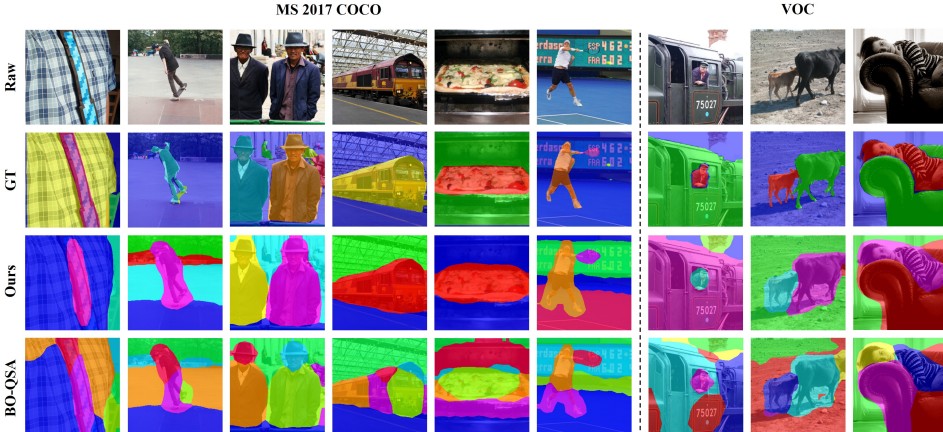

Figure 2: Qualitative results show that MetaSlot's dynamic slot allocation mitigates BO-QSA's over-segmentation, such as splitting a train into unrelated parts, due to its fixed slot count.

generative modules. However, these methods remain constrained by a fixed number of slots, resulting in persistent over-segmentation issues. FT-DINOSAUR [47] mitigates redundancy by selecting the top $k$ most probable slots during the decoding phase, and SOLV [52] clusters aggregated slots to improve semantic consistency. Nonetheless, these approaches rely on heuristic, non-learning-based strategies with explicit thresholding during decoding. While AdaSlot [53] directly predicts the number of slots from feature maps, it does not achieve quantitative improvements over standard Slot Attention on real-world datasets. Furthermore, none of these existing methods incorporate explicit object priors or semantic cues during initialization.

On the theoretical front, a number of studies have offered formal interpretations of object-centric learning (OCL) [54, 27, 55–58]. In terms of evaluation methodologies, recent works have examined the generalization ability of object-centric representations across various downstream tasks, including visual question answering (VQA) [59], world modeling [24, 25, 60, 61], and video generation [59, 62]. Furthermore, several studies have investigated the robustness of object-centric models under out-of-distribution (OOD) conditions [63, 64, 47]. Other works have explored the use of vector quantization mechanisms to improve object disentanglement and interpretability. For instance, methods such as [65, 66] learn hierarchical, compositional discrete representations that align with objects and their attributes.

## 4   Experiments

Overall, our study targets two goals: first, to show that MetaSlot–when substituted for vanilla slot attention–consistently enhances the performance of object-centric learning (OCL) models; and second, to demonstrate that MetaSlot plugs in naturally to both Transformer-based and diffusion-based OCL frameworks, underscoring its broad applicability. We evaluate its impact on two canonical tasks: object discovery, which demands pixel-accurate masks for every object instance, and set prediction, where classification accuracy reveals how much object information the slots capture and thus reflects the quality of the learned representations.

**Datasets**   We include both synthetic and real-world datasets. ClevrTex [67] comprises synthetic images, each with about 10 geometric objects scattered in complex backgrounds. MS COCO 2017 [68] is a recognized real-world image dataset, and we use its challenging panoptic segmentation and instance-level object annotations. PASCAL VOC 2012 [69] is a real-world image dataset, and we use its instance segmentation. We also report results on the real-world video dataset HQ-YTVIS [70], which contains large-scale short videos from YouTube.

**Training Details**   To eliminate confounding implementation differences, we re-implemented every baseline from scratch rather than reusing published results. All experiments share identical data augmentation pipelines and use the DINOv2 ViT(s/14) [50] as the OCL encoder, with matched

training hyperparameters. Every model—including both the baselines and our variants augmented with MetaSlot–was trained for 50 k steps with the Adam optimizer [71] on a single NVIDIA V100 GPU using 16-bit mixed precision and a batch size of 32; the MetaSlot codebook size was fixed to 512 throughout. As the most advanced publicly available aggregator currently surpassing vanilla Slot Attention, BO-QSA [29] is adopted as the default module in all baselines. This uniform setup ensures fair and reproducible comparisons, enabling precise evaluation of MetaSlot's contribution. All reported results are averaged over three random seeds to mitigate stochastic variance.

## 4.1 Evaluate on Object Discovery

**Models** We integrate MetaSlot into object-centric learning (OCL) frameworks and systematically benchmark it against a range of classic models (Table 1) as well as state-of-the-art models (Table 2) to highlight its performance gains. Concretely, SLATE [22] employs a Transformer decoder for autoregressive reconstruction. DINOSAUR [46] uses an MLP-based hybrid decoder to reconstruct directly in DINO feature space, while VideoSAUR [72] adapts this design to video. SlotDiffusion [42] performs decoding with a conditional diffusion model, and SPOT [73] combines nine permutation-based Transformer decoders with a self-training strategy. Evaluating MetaSlot within each of these heterogeneous decoding paradigms enables a comprehensive assessment of its versatility. We exclude IODINE [49], ISA [74], SAVi [21], SAVi++ [37], and MoTok [75] due to outdated performance or reliance on multi-modal priors that hinder fair comparison under our unified setting. Appendix E presents a comparative evaluation of MetaSlot against AdaSlot[53], SlotContrast[76], SysBinder[65], and NLoTM[66] across multiple datasets, further demonstrating the effectiveness of MetaSlot.

**Metrics** The object-discovery task provides a straightforward view of how effectively individual slots separate distinct objects. Following standard practice in OCL research, we assess representation quality by comparing the mask assigned to each slot with the instance-level ground-truth masks. Concretely, we report the Adjusted Rand Index (ARI) and Foreground Adjusted Rand Index (FG-ARI) [77] to measure clustering similarity, and evaluate mean Intersection-over-Union (mIoU) and mean Best Overlap (mBO) to quantify how well the discovered masks align with the real objects.

**Analysis** To comprehensively evaluate the effectiveness of the proposed MetaSlot aggregator, we integrate it into several mainstream OCL decoding frameworks (MetaSlot$_{Mlp}$, MetaSlot$_{Tfd}$, MetaSlot$_{Dfz}$) and directly compare it against their original implementations (DINOSAUR, SLATE, SlotDiffusion).

As shown in Table 1, MetaSlot consistently outperforms its corresponding baseline across all decoding frameworks. In the MLP setting, MetaSlot$_{Mlp}$ yields higher decoding accuracy and better reconstruction quality than DINOSAUR. Under the autoregressive setting, MetaSlot$_{Tfd}$ achieves substantial performance gains over the original SLATE. Similarly, in the diffusion-based framework, MetaSlot$_{Dfz}$ consistently surpasses SlotDiffusion. These results demonstrate MetaSlot's strong compatibility and generalization ability across diverse decoder architectures, highlighting its robustness and superiority in complex visual reconstruction tasks. We also visualize the object-segmentation results of MetaSlot$_{Mlp}$ on the COCO and VOC datasets in Fig.2. The examples show that MetaSlot performs dynamic slot allocation effectively, eliminating the over-segmentation problem that afflicts the DINOSAUR baseline, whose BO-QSA [29] aggregator enforces a fixed number of slots.

Furthermore, as shown in Table 2, we compare MetaSlot with recent state-of-the-art methods. Because our goal is not to challenge the entire model architecture but to isolate the impact of the aggregator module. Therefore, in comparing with SPOT [73], MetaSlot$_{Tfd9}$ is trained using only the decoder from SPOT in a single training round, without adopting the two-stage self-distillation strategy proposed in the original work. Remarkably, even under this simplified training setup, our method achieves comparable or even superior performance across all evaluation metrics. Similarly, when comparing with VideoSAUR on the YTVIS(HQ) dataset, simply replacing its aggregator with MetaSlot yields substantial performance improvements–particularly in FG-ARI (+18.3) and mBO (+2.9). These results highlight MetaSlot's adaptability, proving effective in both static and video-level object discovery tasks.

## 4.2 Evaluate on Set Prediction

The set prediction task explicitly reveals the effectiveness of each slot in capturing object information. Following this work [46], images from the MS COCO 2017 dataset are encoded into object-centric

Table 1: Object discovery performance with DINOv2 ViT (s/14) for OCL encoding. The input resolution is 256×256 (224×224). Tfd, MLP and Dfz are Transformer, MLP, and Diffusion [78] for OCL decoding respectively.

| | ClevrTex #slot=11 | | | | COCO #slot=7 | | | | VOC #slot=6 | | | |
|---|---|---|---|---|---|---|---|---|---|---|---|---|
| | ARI | FG-ARI | mBO | mIoU | ARI | FG-ARI | mBO | mIoU | ARI | FG-ARI | mBO | mIoU |
| SLATE | $17.4_{\pm2.9}$ | $87.4_{\pm1.7}$ | $44.5_{\pm2.2}$ | $43.3_{\pm2.4}$ | $17.5_{\pm0.6}$ | $28.8_{\pm0.3}$ | $26.8_{\pm0.3}$ | $25.4_{\pm0.3}$ | $18.6_{\pm0.1}$ | $26.2_{\pm0.8}$ | $37.2_{\pm0.5}$ | $36.1_{\pm0.4}$ |
| MetaSlot$_\text{Tfd}$ | $\mathbf{40.9_{\pm1.7}}$ | $\mathbf{92.4_{\pm0.7}}$ | $\mathbf{49.3_{\pm0.8}}$ | $\mathbf{48.8_{\pm1.6}}$ | $\mathbf{18.6_{\pm1.2}}$ | $\mathbf{33.5_{\pm1.0}}$ | $\mathbf{28.2_{\pm0.7}}$ | $\mathbf{26.7_{\pm0.6}}$ | $\mathbf{20.4_{\pm0.5}}$ | $\mathbf{30.7_{\pm0.8}}$ | $\mathbf{39.0_{\pm0.3}}$ | $\mathbf{37.8_{\pm0.5}}$ |
| DINOSAUR | $50.7_{\pm24.1}$ | $89.4_{\pm0.3}$ | $53.3_{\pm5.0}$ | $52.8_{\pm5.2}$ | $18.2_{\pm1.0}$ | $35.0_{\pm1.2}$ | $28.3_{\pm0.5}$ | $26.9_{\pm0.5}$ | $21.5_{\pm0.7}$ | $35.2_{\pm1.3}$ | $40.6_{\pm0.6}$ | $39.7_{\pm0.6}$ |
| MetaSlot$_\text{Mlp}$ | $\mathbf{64.6_{\pm0.3}}$ | $\mathbf{89.6_{\pm0.4}}$ | $\mathbf{55.2_{\pm0.5}}$ | $\mathbf{54.5_{\pm0.6}}$ | $\mathbf{22.4_{\pm0.3}}$ | $\mathbf{40.3_{\pm0.5}}$ | $\mathbf{29.5_{\pm0.2}}$ | $\mathbf{27.9_{\pm0.2}}$ | $\mathbf{32.2_{\pm1.2}}$ | $\mathbf{43.3_{\pm0.7}}$ | $\mathbf{43.9_{\pm0.3}}$ | $\mathbf{42.1_{\pm0.2}}$ |
| SlotDiffusion | $66.1_{\pm1.3}$ | $\mathbf{82.7_{\pm1.6}}$ | $54.3_{\pm0.5}$ | $53.4_{\pm0.8}$ | $17.7_{\pm0.5}$ | $28.7_{\pm0.1}$ | $27.0_{\pm0.4}$ | $25.6_{\pm0.4}$ | $17.0_{\pm1.2}$ | $21.7_{\pm1.8}$ | $35.2_{\pm0.9}$ | $34.0_{\pm1.0}$ |
| MetaSlot$_\text{Dfz}$ | $\mathbf{81.9_{\pm0.2}}$ | $77.6_{\pm1.1}$ | $\mathbf{64.2_{\pm0.6}}$ | $\mathbf{62.8_{\pm0.7}}$ | $\mathbf{17.7_{\pm0.2}}$ | $\mathbf{32.2_{\pm0.7}}$ | $\mathbf{27.2_{\pm0.2}}$ | $\mathbf{25.8_{\pm0.1}}$ | $\mathbf{19.0_{\pm0.3}}$ | $\mathbf{24.1_{\pm0.3}}$ | $\mathbf{36.5_{\pm0.1}}$ | $\mathbf{35.3_{\pm0.1}}$ |

Table 2: Comparison with SOTA methods: SPOT on MS COCO 2017 (images) and VideoSAUR on YTVIS-HQ (videos). All models use a DINOv2 ViT(s/14) backbone. The input resolution is 256×256 (224×224).

| | COCO #slot=7 | | | | | YTVIS(HQ) #slot=7 | | | |
|---|---|---|---|---|---|---|---|---|---|
| | ARI | FG-ARI | mBO | mIoU | | ARI | FG-ARI | mBO | mIoU |
| SPOT | $20.3_{\pm0.7}$ | $41.1_{\pm0.3}$ | $30.4_{\pm0.1}$ | $\mathbf{29.0_{\pm0.9}}$ | VideoSAUR | $33.0_{\pm0.6}$ | $49.0_{\pm0.9}$ | $30.8_{\pm0.4}$ | $\mathbf{30.1_{\pm0.6}}$ |
| MetaSlot$_\text{Tfd9}$ | $\mathbf{23.1_{\pm0.2}}$ | $\mathbf{41.2_{\pm0.3}}$ | $\mathbf{30.5_{\pm0.3}}$ | $28.6_{\pm0.8}$ | MetaSlot-VideoSAUR | $\mathbf{60.0_{\pm2.3}}$ | $\mathbf{67.3_{\pm2.1}}$ | $\mathbf{33.7_{\pm0.8}}$ | $28.3_{\pm0.7}$ |

representations using OCL. Each slot is tasked with predicting the object category labels and bounding box coordinates via a small MLP. We evaluate classification performance using top-1 accuracy of the category labels and assess regression performance using the $R^2$ score of the bounding box coordinates.

Table 3 shows that the proposed MetaSlot (i.e., MetaSlot$_{Mlp}$) consistently outperforms the baseline [46] in both object classification and bounding box regression tasks. These results indicate that the object representations captured by MetaSlot are superior, effectively improving the encoding of both categorical and spatial information.

Table 3: Set prediction performance

| COCO | class labels | bounding boxes |
|---|---|---|
| #slot=7 | top1↑ | top2↑ |
| DINOSAUR + MLP | 0.33 | 0.54 |
| MetaSlot$_\text{MLP}$ + MLP | **0.36** | **0.56** |

Table 4: Aggregator comparison.

| | COCO #slot=7 | | | |
|---|---|---|---|---|
| | ARI | FG-ARI | mBO | mIoU |
| Slot Attention | $17.2_{\pm0.2}$ | $38.6_{\pm0.6}$ | $27.7_{\pm0.4}$ | $26.5_{\pm0.3}$ |
| BO-QSA | $18.2_{\pm1.0}$ | $35.0_{\pm1.2}$ | $28.3_{\pm0.5}$ | $26.9_{\pm0.5}$ |
| MetaSlot | $\mathbf{22.4_{\pm0.3}}$ | $\mathbf{40.3_{\pm0.5}}$ | $\mathbf{29.5_{\pm0.2}}$ | $\mathbf{27.9_{\pm0.2}}$ |

## 4.3 Interpretability Analysis

Kori et al. [55] interpret Slot Attention (SA) [11] as a Gaussian Mixture Model (GMM), where each slot acts as a Gaussian component explaining a subset of pixels. Building on this view, we shift the focus from concrete objects to abstract prototypes, positing that the real-world distribution can likewise be factorized into a mixture of such prototypes. Concrete objects in the feature map can then be regarded as samples or projections from these prototype distributions. In slot-attention-based object-centric learning, matching these abstract prototypes manifests as slot initialization, while the iterative attention updates project and refine the prototype distributions onto concrete visual appearances.

In SA, all slots are sampled from a shared Gaussian prior, which lacks object-level inductive cues during initialization. BO-QSA [29] mitigates this limitation by assigning independent Gaussian priors to each slot, thereby promoting greater diversity and improving object-attribute binding. However, its fixed slot count limits its adaptability to the diverse objects encountered in real-world scenes. To address this, our proposed MetaSlot introduces a set of adaptive prototype slots that capture abstract representations of real-world entities, further enhancing object binding and improving flexibility in complex scenes.

As shown in Fig. 3, MetaSlot's prototype-based initialization yields slots with pronounced semantic consistency and strong object binding—e.g., all slots from prototype 268 correspond to "keyboard"

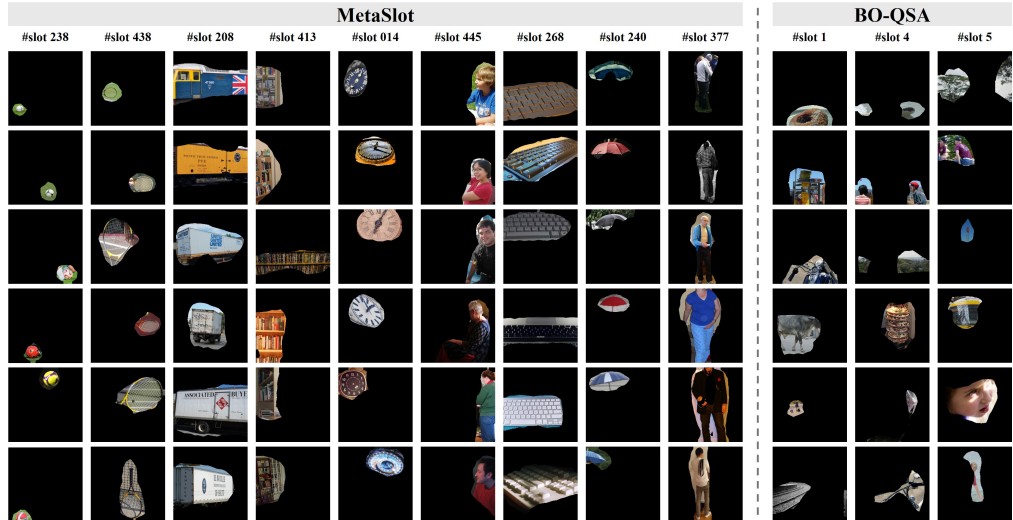

Figure 3: We visualize slot representations initialized from different prototype slots on the COCO dataset, where each column shows a specific initialization slot index–prototype slots in MetaSlot (out of 512) and fixed slots in BO-QSA [29]. MetaSlot's prototype-based initialization yields slots with strong semantic consistency and object binding (e.g., #slot 208 for trucks, #slot 445 for persons). By comparison, the fixed slots in BO-QSA frequently lack such coherent semantic grouping.

objects, while those from prototype 240 correspond to "umbrella" objects. In contrast, BO-QSA, limited by its fixed number of slots, struggles to achieve such fine-grained prototype binding. Additional results on the VOC dataset are reported in Appendix B. Quantitative comparisons under the DINOSAUR [46] decoding framework (Table 4) further confirm that MetaSlot surpasses both SA and BO-QSA on all object discovery metrics, highlighting its superiority in producing disentangled and interpretable slot representations.

## 4.4 Ablations

To assess the effectiveness of the key architectural components of MetaSlot, we perform a comprehensive ablation study on the MS COCO 2017 dataset. All experiments adopt the DINOSAUR framework with a DINOv2 ViT (s/14) encoder and fix the number of slots to seven. To validate the contribution of individual design choices, we further evaluate two ablated variants: 'MetaSlot w/o noise' omits progressively attenuated noise during slot updates. As shown in Fig.4, injecting noise leads to a lower Adjusted Rand Index (ARI, left) and a higher mean best overlap (mBO, right), implying faster and more stable slot aggregation. 'MetaSlot w/o mask' disables the prototype-based masking strategy. Table 5 summarizes performance across three metrics–Foreground ARI (FG-ARI), mean best overlap (mBO), and mean Intersection-over-Union (mIoU), indicating that each component is pivotal for precise slot-to-object alignment and effective spatial disentanglement. In addition, Appendix E presents an analysis of the codebook prototype size, where we observe that increasing the number of prototypes yields only marginal improvements in performance. We also compare MetaSlot models with varying slot counts, and the empirically optimal number of slots aligns with the long-established consensus within the object-centric learning community. Furthermore, we include additional ablations on architectural components, which provide further evidence for the robustness and effectiveness of our model design.

## 5 Conclusion

This paper introduces MetaSlot, a novel aggregator for object-centric learning (OCL) that addresses two long-standing limitations of conventional Slot Attention models: the fixed number of slots and reliance on random initialization. MetaSlot incorporates a global vector-quantized (VQ) prototype codebook alongside a two-stage aggregate-and-deduplicate framework. This design enables the model to adaptively adjust the number of slots based on scene complexity and to initialize slots

Table 5: Ablation study on architectural components. Backbone: DINOv2 ViT (s/14).

| | COCO #slot=7 | | |
| --- | --- | --- | --- |
| | FG-ARI | mBO | mIoU |
| MetaSlot w/o noise | $39.4_{\pm 0.3}$ | $28.9_{\pm 0.4}$ | $27.4_{\pm 0.4}$ |
| MetaSlot w/o mask | $38.2_{\pm 0.2}$ | $28.5_{\pm 0.1}$ | $26.9_{\pm 0.3}$ |
| MetaSlot | $\mathbf{40.3_{\pm 0.5}}$ | $\mathbf{29.5_{\pm 0.2}}$ | $\mathbf{27.9_{\pm 0.2}}$ |

Figure 4: Training curves for MetaSlot$_{\text{Mlp}}$ with/without progressively attenuated noise.

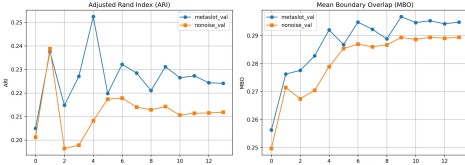

with semantically meaningful object representations. Extensive experiments show that MetaSlot consistently achieves substantial gains across a range of OCL tasks and offers a robust foundation for future research in OCL and its downstream applications.

# 6 Limitations and Future Directions

Despite its promising results, MetaSlot still faces several limitations that point to fruitful directions for future research. First, the use of absolute positional encoding inherited from the original Slot Attention makes the model non-equivariant to image translations, potentially causing the codebook prototypes to capture position-dependent noise patterns. Future work could leverage translation-equivariant mechanisms such as ISA [74] to promote the emergence of consistent, position-agnostic representations. Second, current object prototypes primarily encode global shape or semantic information while overlooking finer-grained attribute compositions. Enhancing prototype optimization through compositional generalization—that is, constructing compound prototypes integrating multiple attribute-level features—may yield richer and more discriminative object cues for downstream tasks. Third, owing to MetaSlot's two-stage and iterative optimization design, the framework inherently contains self-supervisory signals. Exploring ways to identify the semantically complete object slots emerging in the second stage and using them to provide weak supervision for the aggregation process in the first stage could further advance the development of variable-slot object-centric learning. Finally, MetaSlot has yet to be extensively evaluated under out-of-distribution (OOD) conditions. Systematic studies across diverse domains, coupled with efforts to better align the learned codebook prototype distributions with real-world object distributions, could further improve the model's cross-sample and cross-domain generalization.

## Acknowledgments and Disclosure of Funding

We acknowledge the support of Finnish Center for Artificial Intelligence (FCAI), Research Council of Finland flagship program. We thank the Research Council of Finland for funding the projects ADEREHA (grant no. 353198). We also appreciate CSC - IT Center for Science, Finland, for granting access to supercomputers Mahti and Puhti, as well as LUMI, owned by the European High Performance Computing Joint Undertaking (EuroHPC JU) and hosted by CSC Finland in collaboration with the LUMI consortium. Furthermore, we acknowledge the computational resources provided by the Aalto Science-IT project through the Triton cluster.

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

## A  Method

**Algorithm 1:** MetaSlot.

**1** ___

**Input:** input features $input$, learnable queries $init$, number of iterations $T$
**Output:** object-centric representation $S^{final}$

**2 Modules:** stop gradient module $\text{SG}(\cdot)$, slot attention module $\text{SA}(\cdot, \cdot)$, masked slot attention module $\text{MSA}(\cdot, \cdot, \cdot)$, vector quantization $\text{VQ}(\cdot)$, prune duplicate slots module $\text{Prune}(\cdot)$, update prototype codebook module $\text{Update}(\cdot, \cdot)$, noise injection module $\text{Noisy}(\cdot, \cdot)$

**3** ___

**4 # First-Stage Aggregation :**

**5** $S^{mid} \leftarrow init.detach()$;
**6 for** $t = 1$ **to** $T$ **do**
**7** $\quad \lfloor \; S^{mid} \leftarrow \text{SA}(S^{mid}, inputs)$;
**8** $\hat{S}, idx \leftarrow \text{VQ}(S^{mid})$;
**9** $\hat{S}_{mask}, mask \leftarrow \text{Prune}(\hat{S}, idx)$;

**10 # Second-Stage Aggregation :**

**11** $S^{final} \leftarrow \hat{S}_{mask}$;
**12 for** $t = 1$ **to** $T - 1$ **do**
**13** $\quad \mid \; input_{noisy} \leftarrow \text{Noisy}(input, t)$;
**14** $\quad \lfloor \; S^{final} \leftarrow \text{MSA}(S^{final}, input_{noisy}, mask)$;
**15** $S^{final} \leftarrow \text{SG}(S^{final}) + init - \text{SG}(init)$;
**16** $input_{noisy} \leftarrow \text{Noisy}(input, T)$;
**17** $S^{final} \leftarrow \text{MSA}(S^{final}, input_{noisy}, mask)$;

**18 # Prototype update :**

**19** $\text{Update}(\text{SG}(S^{final}), mask)$;
**20 return** $S^{final}$

**21** ___

## B  Visualization of prototype slots

As shown in Fig. 5, we visualize the refined slots $S^{\text{final}}$ corresponding to the initialization prototype slots on the VOC dataset. The codebook contains 512 object prototypes in total. However, since the VOC-trained model is relatively limited in object diversity and scale, many prototype slots receive few or no refined slots assigned to them, making comprehensive visualization infeasible. Therefore, for practical and technical reasons, we focus on the top 20 most active prototype slots—defined as those associated with the largest number of refined slots in the model trained on the VOC dataset. For each selected prototype slot, we randomly sample six refined slots from its assigned set and visualize their corresponding image patches. As shown in the figure, the results clearly demonstrate that the prototype slots exhibit strong concept binding behavior, with refined slots consistently aligned to semantically coherent object categories. These findings are consistent with our theoretical expectations regarding the semantic consistency induced by prototype-guided initialization.

## C  Experimental Details

**Implementation and Reproducibility.**    To ensure a fair comparison, we re-implemented all baseline models from scratch rather than relying on publicly reported results. Throughout all experiments, we kept data augmentation strategies, the visual feed-forward module (VFM) in the OCL encoder—based on DINOv2 ViT-s/14 [50]—and all training hyperparameters identical to those reported in the original papers. Furthermore, we replaced each model's original variational autoencoder (VAE) component with a large-scale pre-trained TAESD module [78], which is based on Stable Diffusion.

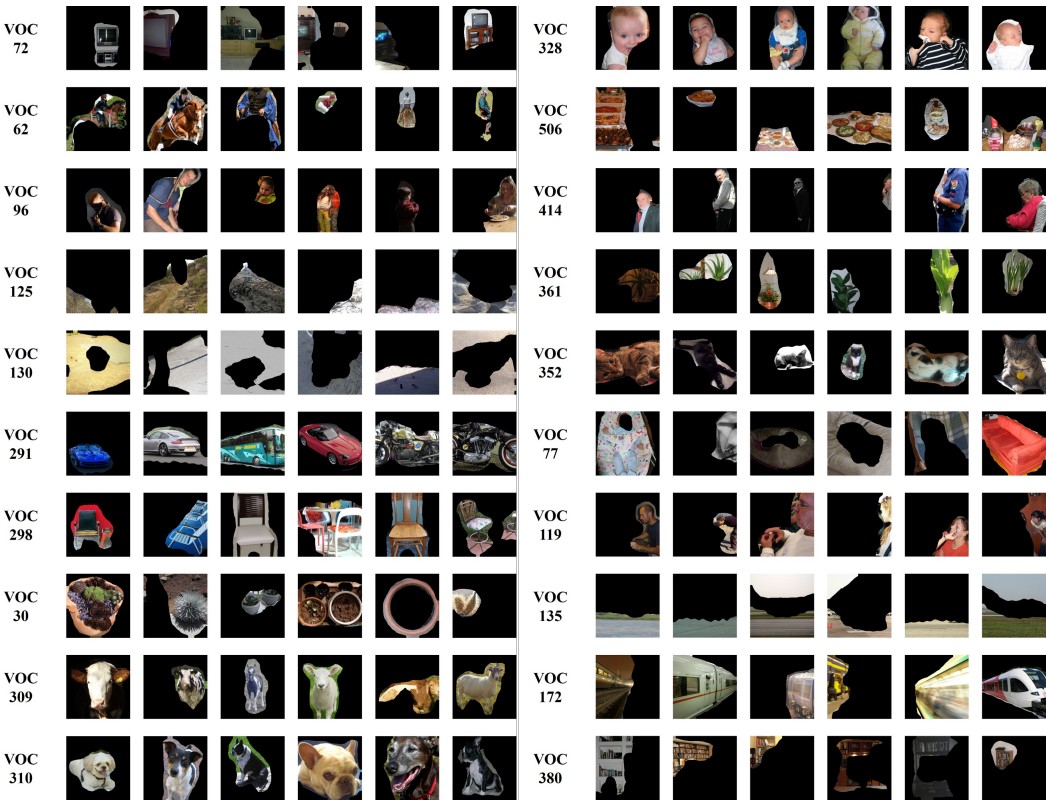

Figure 5: Visualize slot representations initialized from different prototype slots on the VOC dataset.

All models—including both the MetaSlot-augmented variants and their respective baselines—were trained using the Adam optimizer [71] on a single NVIDIA V100 GPU with 16-bit mixed precision. Each run consisted of 50000 training steps, with a batch size of 32 and four data-loading workers. We set the initial learning rate to $2 \times 10^{-4}$ and maintained it throughout training. For the MetaSlot module, the codebook size was fixed at 512, the feature map resolution at $256 \times 256$, and the embedding dimension at 256. The number of slots for each model remained consistent with its original baseline configuration. To reduce the impact of randomness, all experiments were repeated with three different random seeds, and we report the mean results.

**Pre-trained VQ-VAE Configuration for SLATE and Slot Diffusion.** For the SLATE and Slot Diffusion baselines, we adhered to the standard VQ-VAE implementation based on ResNet-18. Specifically, we used a codebook size of 4096 and an embedding dimension of 256. Pre-training was conducted for 30000 steps with a batch size of 64 shared between the text and vision branches, four data-loading workers, and an initial learning rate of $2 \times 10^{-3}$. The feature map resolution remained $256 \times 256$. As before, each configuration was evaluated over three independent runs, and the reported metrics represent the averaged performance across these trials.

## D   Supplementary Ablation Experiments

**Impact of Codebook Size** As shown in table 6, we investigated the impact of different codebook sizes (256, 512, and 1024) on model performance. We observed that codebook size has a limited effect on performance; however, a larger prototype set allows for finer distinctions between slots that share semantic concepts but differ in spatial location. For example, as shown in Fig. 3, slot #445 corresponds to the concept of "person" located on the right side of the image, while slot #337 represents the same concept but appears at the center of the scene.

**Impact of Slot Number** As shown in table 7, we examined how the number of slots affects final performance. Since the codebook and the aggregator module are jointly optimized, the quality of the

Table 6: Results of MetaSlot with varying codebook sizes on MS COCO.

| | DINOSAUR on MS COCO | | | |
|---|---|---|---|---|
| | ARI | FG-ARI | mBO | mIoU |
| MetaSlot, codebook_size=256 | **26.7** | 38.1 | 29.4 | 27.5 |
| MetaSlot, codebook_size=512 | 22.4 | **40.3** | **29.5** | **27.9** |
| MetaSlot, codebook_size=1024 | 20.0 | 40.4 | 29.2 | 27.3 |

Table 7: Results of MetaSlot with varying slot numbers on MS COCO.

| | DINOSAUR on MS COCO | | | |
|---|---|---|---|---|
| | ARI | FG-ARI | mBO | mIoU |
| MetaSlot, slot_num=5 | **25.5** | 37.8 | 29.3 | 27.4 |
| MetaSlot, slot_num=7 | 22.4 | **40.3** | **29.5** | **27.9** |
| MetaSlot, slot_num=11 | 19.8 | 38.7 | 28.4 | 26.9 |
| MetaSlot, slot_num=15 | 15.9 | 36.0 | 27.2 | 26.0 |

aggregator—particularly in early training stages—can significantly influence codebook convergence. We found that the empirically optimal slot count aligns well with long-standing choices commonly adopted in the community.

**Impact of Architectural Components**  As shown in table 8, we conducted additional experiments to gain deeper insight into the working mechanism of MetaSlot. Specifically, we tested two variants: (1) removing the prototype guidance and instead initializing slots in the second stage by sampling from separate Gaussian distributions, as done in the first stage; and (2) disabling the reactivation mechanism for stale (dead) prototypes during the codebook update process. The results confirm the effectiveness of our module design and are consistent with the theoretical assumptions proposed in the paper.

Furthermore, prior studies in object-centric learning have consistently shown that simply increasing the number of Slot Attention iterations does not lead to meaningful performance gains. To validate this observation, we performed an additional ablation study on the BO-QSA model by increasing its iteration count to 6, matching that of MetaSlot. As shown in the results, increasing the iteration count alone does not improve BO-QSA's performance, further highlighting the importance of our design choices beyond iteration depth.

Table 8: Supplementary ablation on architectural components.

| | MS COCO #slot=7 | | |
|---|---|---|---|
| | FG-ARI | mBO | mIoU |
| MetaSlot w/o proto | 40.2 | 29.1 | 27.6 |
| MetaSlot w/o prune | 40.0 | 29.1 | 27.5 |
| MetaSlot | **40.3** | **29.5** | **27.9** |
| BO-QSA,iter_num=6 | 37.9 | 27.7 | 26.3 |

Table 9: Comparison with SlotContrast on the MOVi-C dataset.

| Method | FG-ARI | mBO |
|---|---|---|
| SlotContrast | 62.4 | 30.6 |
| MetaSlot | **63.9** | **35.0** |

# E  Supplementary Comparative Experiments

**Comparative Evaluation against SlotContrast**  As shown in table 10, we further include a comparison with the SlotContrast[76] model on the MOVi-C dataset. In these experiments, MetaSlot does not employ SlotContrast losses or any temporal-specific enhancements. Due to constraints in time and computational resources, we used a batch size of 8 (vs. 64 in SlotContrast), 24 frames per sample, and trained for up to 20,000 steps (vs. 100,000 in SlotContrast). Despite these limitations, MetaSlot still demonstrates notable performance advantages.

It is worth noting that SlotContrast's contrastive loss was originally designed for a fixed number of slots, and extending it to handle variable slot counts would require additional effort. Nevertheless, as discussed earlier, even without incorporating SlotContrast's core contribution—the contrastive

loss—MetaSlot achieves comparable or even superior performance in object discovery tasks. We believe that integrating SlotContrast's contrastive learning objective in the first stage with MetaSlot's dynamic aggregation in the second stage could further enhance performance, particularly for temporal object discovery. However, this would necessitate architectural modifications that are beyond the scope of the current work and are left for future research.

Table 10: Comparison with SlotContrast on the MOVi-C dataset.

| Method | FG-ARI | mBO |
|---|---|---|
| SlotContrast | 62.4 | 30.6 |
| MetaSlot | **63.9** | **35.0** |

Table 11: Comparison with AdaSlot on the MS COCO dataset.

| Method | FG-ARI | mBO |
|---|---|---|
| AdaSlot | 35.6 | 29.4 |
| BO-QSA | 35.0 | 28.3 |
| MetaSlot | **40.3** | **29.5** |

**Comparative Evaluation against AdaSlot**   As shown in Table 11, we further report the results of AdaSlot[53] compared with MetaSlot. All models use a DINOv2 ViT-S/14 backbone, and the input resolution is 256×256 (or 224×224). We speculate that the performance difference is partly due to the fact that, when the number of slots varies, the model can no longer apply separate Gaussian initialization to each slot individually.

**Additional Evaluation of VQ-based Object-Centric Methods**   We further evaluated several object-centric learning methods that employ vector quantization. In SysBinder[65] and NLoTM[66], the quantization mechanisms operate at the attribute level within each slot. However, this design does not necessarily lead to improved representation quality in unsupervised settings. In synthetic datasets, object attributes such as color, size, shape, and material are relatively fixed and can often be described using fewer than ten shared labels. In contrast, real-world datasets such as COCO and VOC contain objects with far more diverse and non-shared attributes. For instance, the object *person* cannot be meaningfully described with the same attribute set as the object *mouse*. This discrepancy likely explains why SysBinder and NLoTM were not originally evaluated on real-world benchmarks.

Furthermore, the initial intuition behind MetaSlot is inspired by Platonic philosophy. In Plato's theory of Forms, sensible particulars are intelligible only insofar as they "participate" (metechē) in a transcendent Form (Phaedo 100c–d). Similarly, Aristotle maintains in the Categories that accidents belong to substances only as predicates of discourse, not as essential ingredients shared by all beings (Categories 2a11–19).

Our empirical evaluation on COCO and ClevrTex shows that MetaSlot substantially outperforms both SysBinder and NLoTM. Although SysBinder achieves a slight advantage in FG-ARI on ClevrTex, it underperforms the DINOSAUR baseline on other key metrics such as ARI, mBO, and mIoU. Moreover, the SysBinder paper only reports FG-ARI and does not include ARI, mBO, or mIoU—metrics widely regarded as standard for evaluating object discovery. All models use a DINOv2 ViT-S/14 backbone, and the input resolution is 256×256 (or 224×224).

Table 12: Comparative results of SysBinder, NLoTM, DINOSAUR, and MetaSlot on MS COCO.

| | MS COCO | | | |
|---|---|---|---|---|
| | ARI | FG-ARI | mBO | mIoU |
| SysBinder | 16.8 | 37.7 | 27.2 | 26.0 |
| NLoTM | 57.8 | 14.7 | 23.6 | 17.8 |
| DINOSAUR | 18.2 | 35.0 | 28.3 | 26.9 |
| MetaSlot | **22.4** | **40.3** | **29.5** | **27.9** |

Table 13: Comparative results of SysBinder, NLoTM, DINOSAUR, and MetaSlot on ClevrTex.

| | ClevrTex | | | |
|---|---|---|---|---|
| | ARI | FG-ARI | mBO | mIoU |
| SysBinder | 21.7 | **91.4** | 46.8 | 46.5 |
| NLoTM | 21.5 | 88.7 | 44.1 | 43.1 |
| DINOSAUR | 50.7 | 89.4 | 53.3 | 52.8 |
| MetaSlot | **64.6** | 89.6 | **55.2** | **54.5** |

