# OpenReview forum: "MetaSlot: Break Through the Fixed Number of Slots in Object-Centric Learning"
_NeurIPS.cc/2025/Conference — NeurIPS 2025 poster_

### Official Review · Reviewer_gev3 · 2025-06-29

**Clarity:** 3
**Significance:** 2
**Originality:** 2
**Rating:** 4
**Confidence:** 4

**Summary:**

This paper proposes MetaSlot, a variant of Slot Attention bottleneck that utilizes vector quantization to identify the prototypes of slots and eliminate duplicate slots, thereby overcoming the limitation of the fixed number of slots. The authors further propose the second-stage aggregation based on quantized slots and other techniques, like noise annealing, to further improve object discovery results.

**Questions:**

Same as the weaknesses part mentioned above.

I am glad to accept the paper if the author could address my concerns about weakness 2 and weakness 3.

Moreover, there is an additional question:

1. A potential concern arises from the slot deduplication procedure: when multiple objects with similar prototypes appear in an image, how does the method avoid erroneously removing semantically similar objects? For example, the third picture in Figure 2 with two persons in the foreground. The authors are encouraged to visualize the corresponding codebook ID of the two persons and the other slots in the codebook.

**Ethical Concerns:**

["NO or VERY MINOR ethics concerns only"]

**Final Justification:**

The discussion with the authors satisfactorily addressed my main concern regarding the lack of comparisons with other slot quantization methods. Given this, I have increased my final score. No other major issues remain.

**Limitations:**

Yes

**Paper Formatting Concerns:**

No.

**Quality:**

2

**Strengths And Weaknesses:**

**Strengths:**
1. The paper is clearly written and easy to follow. The author proposes a novel two-stage slot variant: the method first quantizes the slots and then refines these quantized slots with noise, resulting in the final aggregated slots. This approach offers a new and reasonable direction for slot quantization research.
2. Notably, the quantized slots can be deduplicated, enabling a dynamic number of slots, as illustrated in Fig. 2. Furthermore, the method outperforms prior work like AdaSlot by achieving improved grouping results on real-world datasets.
3. The author also provides extensive experimental results, evaluating the approach with three different types of decoders.

**Weaknesses:**
1. The authors are encouraged to provide object discovery visualizations on ClevrTex to further illustrate their method's effectiveness in the dynamic slot allocation.
2. Additionally, the comparison in Figure 3 does not sufficiently support the claims of interpretability analysis. Specifically, the slot indices in BO-QSA do not necessarily correspond to clear visual concepts. To address this, the authors should consider following the approach used in SLATE by constructing a visual concept library with K-means clustering on the computed slots after $T$ iterations.
3. The related work section also overlooks a substantial body of literature on slot-related vector quantization (VQ), such as SysBinder. Since incorporating the quantization module could naturally enhance the slot attention mechanism, it would be valuable for the authors to include comparisons with methods like SysBinder[1], COSA[2] and NLoTM [3].

[1] Singh, Gautam, Yeongbin Kim, and Sungjin Ahn. "Neural systematic binder." ICLR 2023

[2] Kori, Avinash, et al. "Grounded Object-Centric Learning."   ICLR 2024

[3] Wu, Yi-Fu, Minseung Lee, and Sungjin Ahn. "Neural language of thought models."  ICLR 2024.

[4] Wu, Yi-Fu, Minseung Lee, and Sungjin Ahn. "Object-Centric Semantic Vector Quantization."  NeurIPS 2023 Causal Representation Learning Workshop.

[5] Kirilenko, Daniil, et al. "Quantized disentangled representations for object-centric visual tasks." PReMI 2023.

---

> ### Author Rebuttal · Authors · 2025-07-31
>
> We appreciate the reviewer’s constructive feedback and address the concerns as follows.
>
> >  Provide object discovery visualizations on ClevrTex
>
> As visualizations cannot be included during the rebuttal stage, we will add a dedicated section in the appendix of the revised paper to present Figure 7 (visualization results on the ClevrTex dataset) and provide a detailed discussion of this phenomenon.
>
> In ClevrTex’s simple and consistent scenes, we observe that merely adding noise annealing to Slot Attention enables the model to bind each object to a distinct slot while automatically leaving redundant slots empty. Specifically, without noise annealing, the model often splits the background across multiple slots, whereas with noise annealing, the background consistently remains confined to a single slot.
>
> Similarly, in the Clevr dataset, shadows caused by lighting often lead the original Slot Attention to mistakenly assign parts of the background to object slots. In contrast, noise annealing helps maintain clean and exclusive slots for objects.
>
> > The slot indices in BO-QSA do not necessarily correspond to clear visual concepts. To address this, the authors should consider following the approach used in SLATE by constructing a visual concept library with K-means clustering on the computed slots after $T$ iterations.
>
> In this section, we would like to clarify that the ability of slot initialization to endow slots with concept-binding capabilities is not the central focus of this paper. Rather, this argument originates from the appendix of the BO-QSA paper, upon which MetaSlot builds and extends. In the BO-QSA appendix, visualizations on single-category datasets such as CUB-200 Birds and Stanford Dogs are presented to illustrate concept binding without the need for clustering. These visualizations clearly show that distinct visual concepts (e.g., bird body, background, tree trunk, leaves) are consistently assigned to separate slots, and that the slot binding order remains stable across the dataset. Notably, in datasets with prominent background elements, a dedicated slot consistently captures the background throughout the entire dataset.
>
> However, in more complex and diverse real-world datasets such as COCO and VOC, where a wide range of object categories exist, the visual representations do not easily reveal clear semantic groupings. To address this, we employed t-SNE clustering to visualize the distribution of BO-QSA and MetaSlot’s prior slots within the posterior slots. The results show that the priors provide good coverage of the posterior distribution and are highly similar to the resulting object-level posterior slots.
>
> As we are unable to present images during the rebuttal phase, we kindly refer the reviewer to Figures 10, 11, and 12 in the appendix of the BO-QSA paper. Additionally, in the revised version, we will include visual comparisons of ObjectRoom generated using both BO-QSA and the original Slot Attention, as well as the corresponding t-SNE clustering visualizations, in the appendix.
>
> Moreover, you have accurately captured the essence of our object concept library construction. While SLATE applies k-means clustering as a post-processing step—grouping slots from the entire dataset into a more compact prototype set after training—MetaSlot integrates this process directly into training via a vector quantization (VQ) codebook, effectively performing end-to-end k-means clustering over the dataset-wide slot representations. We employ BO-QSA in the first stage to initialize clustering and fully leverage the rich semantics of the prototype library to guide the second-stage aggregation, thereby achieving superior performance. Through careful design, MetaSlot enables end-to-end joint optimization, surpassing post-hoc approaches like SLATE in both elegance and effectiveness.
>
> >It would be valuable for the authors to include comparisons with methods like SysBinder, COSA and NLoTM
>
> Thank you for pointing out these three valuable and interesting works. First, the vector quantization mechanisms used in SysBinder and NLoTM operate at the attribute level within each individual slot. However, such mechanisms do not necessarily guarantee improved representation performance in unsupervised learning. This is because, in synthetic datasets, object attributes are relatively fixed—such as size, color, material, and shape—and all objects in the scene can typically be described using fewer than ten shared attribute labels. In contrast, real-world datasets like COCO and VOC contain objects with infinitely divisible and non-shared attribute information. For example, the object "person" cannot be meaningfully described using the same set of attribute labels as the object "mouse." We believe this is also a primary reason why these methods were not evaluated on real-world datasets.
>
> Furthermore, the initial intuition behind MetaSlot is inspired by Platonic philosophy. In Plato’s theory of Forms, sensible particulars are intelligible only insofar as they “participate” (*metechē*) in a transcendent Form (*Phaedo* 100c–d). Similarly, Aristotle maintains in the *Categories* that accidents belong to substances only as predicates of discourse, not as essential ingredients shared by all beings (*Categories* 2a11–19).
>
> At the practical level, SysBinder uses slots as queries that interact with a prototype memory via cross-attention, enabling soft access to the memory bank. This mechanism differs from standard vector quantization, which typically relies on nearest-neighbor search to select discrete representations. We further provide evaluation results of SysBinder and NLoTM on both the COCO and ClevrTex datasets, showing that MetaSlot significantly outperforms both methods. Although SysBinder show a slight advantage in FG-ARI on ClevrTex, they perform worse than the DINOSAUR baseline on other key metrics such as ARI, MBO, and mIoU. Additionally, the SysBinder paper only reports FG-ARI and does not include ARI, MBO, or mIoU—metrics widely regarded as standard for evaluating object discovery.
>
> The poor performance of SysBinder and NLoTM on COCO further supports the above analysis. Moreover, NLoTM exhibited mode collapse on the COCO dataset. In contrast, MetaSlot outperforms the DINOSAUR baseline across all four evaluation metrics.
>
> As for CoSA, the number of objects is indirectly obtained via spectral decomposition, rather than learned adaptively across the dataset by identifying object prototypes. Moreover, CoSA derives its codebook from prior slots. While this prior can effectively guide the refinement process to form the posterior slot distribution, there is no explicit alignment mechanism between the prior and the posterior. In contrast, MetaSlot updates the prototype slots using the posterior slots produced by MSA, ensuring that the distribution of prototypes remains precisely aligned with that of the posterior slots.
>
> Since CoSA did not release its implementation code and only reported ARI as the evaluation metric—which, compared to FG-ARI and MBO, is less indicative of object discovery performance—we did not include a direct comparison with this method. Finally, we will incorporate a concise version of the above discussion into the Related Work section and add the corresponding experiments to the Experiments section.
>
> | MS COCO   | ARI  | FG-ARI | MBO  | mIoU |
> |----------------|------|--------|------|------|
> | SysBinder |  16.8 | 37.7   | 27.2 | 26.0 |
> | NLoTM | 57.8 | 14.7   | 23.6 | 17.8 |
> | DINOSAUR | 18.2 | 35.0   | 28.3 | 26.9 |
> | MetaSlot | 22.4 | 40.3   | 29.5 | 27.9 |
>
> | CLEVRTEX   | ARI  | FG-ARI | MBO  | mIoU |
> |----------------|------|--------|------|------|
> | SysBinder | 21.7 | 91.4   | 46.8 | 46.5 |
> | NLoTM | 21.5 | 88.7   | 44.1 | 43.1 |
> | DINOSAUR | 50.7 | 89.4   | 53.3 | 52.8 |
> | MetaSlot | 64.6 | 89.6   | 55.2 | 54.5 |
>
> >Visualize the corresponding codebook ID of the two persons and the other slots in the codebook.
>
> Thank you for raising this question—it allows us to further clarify the internal mechanism of MetaSlot. As the current model does not explicitly disentangle spatial information from object representations, we have also acknowledged this limitation in the appendix.
>
> In MetaSlot, **each prototype slot encodes not only an “object concept” but also its associated spatial attributes.** For example, Figure 3 shows that #slot 445 represents the concept of a “person” and is bound to the right side of the image, while #slot 337 also corresponds to “person” but is located at the center of the scene. Similarly, in the third image of Figure 2, the person on the left is mapped to #slot 133 (“person” + scene left), while the person on the right corresponds to #slot 187 (“person” + right side with greater foreground prominence). These examples indicate that the prototype slots naturally encode both semantic and spatial information, rather than representing abstract, spatially disentangled concept vectors.
>
> Due to the constraints of the rebuttal phase, we are unable to include all visualizations here. However, we will provide full examples illustrating the correspondence between prior and posterior prototype slots in our released code to facilitate deeper exploration by the community.
>
> At the same time, this observation highlights an open question we discuss in the Limitations section: **how can we disentangle object identity from spatial information while maintaining dynamic slot allocation and preventing over-segmentation via codebook matching?**
>
> Overall, the key contribution of this work lies in introducing the codebook mechanism to support a variable number of slots and in demonstrating the critical role of semantically rich initialization priors for achieving high-quality object discovery with Slot Attention.

---

> > ### Comment · Reviewer_gev3 · 2025-08-06
> >
> > I have carefully read the authors' rebuttal. The authors addressed my main concern, namely the lack of comparison with previous slot quantization methods, and the experiments demonstrate the effectiveness of their algorithm. The authors are encouraged to include related content in future revisions of the manuscript.
> >
> > As for the visualization in Figure 3, I still find the current experiments unconvincing. I believe it is entirely necessary to construct a library using the K-means algorithm for comparison. The construction of the library is implicitly considered in MetaSlot, while BO-QSA lacks such consideration and requires explicit construction.
> >
> > Taking both factors into account, I have decided to increase my score slightly.

---

> > > ### Author Response · Authors · 2025-08-08
> > >
> > > Thank you very much for your constructive feedback. We are glad that our response has addressed your main concern regarding the comparison with previous slot quantization methods. We will include the visualization comparing with K-means, together with the above discussion on quantization comparison, in the final revised manuscript.

---

> > > ### Author Response · Authors · 2025-08-09
> > >
> > > Dear reviewer gev3,
> > >
> > > I hope this message finds you well. I just wanted to kindly remind you regarding the Mandatory Acknowledgement and final rating for our submission, in case they might have been missed. From the discussion, I understood that you were considering updating the rating, so I simply wanted to ensure there is no oversight on this.
> > >
> > > Thank you again for your time and thoughtful feedback.

---

### Official Review · Reviewer_ayFM · 2025-06-30

**Clarity:** 3
**Significance:** 3
**Originality:** 2
**Rating:** 5
**Confidence:** 4

**Summary:**

The paper proposes an object-centric learning method named MetaSlot.
The method leverages two slot attention modules separated by a vector quantization step that serves as a deduplication step for the initial slots, effectively allowing to adapt the number of slots to the image.
The method is evaluated on several datasets, including COCO, Pascal VOC, and ClevrTex, and compared against several baselines, such as SlotDiffusion, SLATE, and DINOSAUR.

**Questions:**

In figure 2, the predicted masks tend to be bigger than the enclosed objects, for example around the people and the cows.
Is this artifact a result of the training procedure? Can it be reduced? How can the reported scores be so high if the masks are so imprecise?

As it stands, the method makes use of two latent clustering methods: the iterative slot attention, which is similar to soft clustering, and vector quantization, which also clusters the slots from phase 1 into a fixed number of prototypes for deduplication.
With this in mind, can we simply see object-centric methods as a form of clustering? And can this perspective help us simplify existing methods rather than adding complexity?
A discussion on this would be welcome and it could help re-center the contribution of the paper.

Minor points:
- There is some repetition in the first two paragraphs of section 2.3, specifically lines 127-133 and 134-142. I recommend revising the text and consolidating these paragraphs.
- What is the difference between the proposed "masked slot attention" and simply dropping slots that are not used?
  The paper described the "masked slot attention" as a novel contribution, but it does not seem too significant.
- L304: typo "mean boundary overlap" -> "mean best overlap".

**Ethical Concerns:**

["NO or VERY MINOR ethics concerns only"]

**Final Justification:**

My initial score was already positive and the rebuttal addressed my concerns, therefore I recommend acceptance.

**Limitations:**

The authors provide a discussion about the limitation of their work in the appendix.
While I agree with most points, I highlight that the method is only tested on a single backbone, DINOv2 ViT-S, and that the experiments in section 4.2 are even more limited, as they include a single dataset and a single baseline.
Given the variability in performance across datasets, tasks, metrics, backbones, and object-centric methods, the results are not as significant as they could be.

**Paper Formatting Concerns:**

all good

**Quality:**

2

**Strengths And Weaknesses:**

The paper reports positive improvements across various datasets and metrics, demonstrating the effectiveness of the proposed method.
The re-implementation and re-evaluation of several baselines, such as DINOSAUR, SLATE and SlotDiffusion, is a a valuable contribution, though the execution is not without issues, as discussed below.

The main weakness is the limited experimental setup, despite it being deemed "extensive" in the text.
- Missing comparison with AdaSlot, which is mentioned in the related work section as the most prominent method to obtain an adaptive number of slots. Considering the focus of the paper, this is a significant omission.
- In the literature on object-centric methods, it is common to include the MOVI-C and MOVI-E datasets in the evaluation. For example, the cited baselines SPOT and SlotDiffusion include these datasets in their evaluation. While I agree that datasets such as CLEVR and ShapeStacks can be left out because of their simplicity, I think the MOVI datasets should be part of the evaluation.
- Only DINOv2 ViT-S is used as the backbone, leaving out larger backbones and alternative models such as CLIP or MAE. This choice reduces the significance of the results, as it is not clear whether the proposed method would work with other backbones. Also note that the cited baselines use larger backbones, typically ViT-B/16 or ViT-S/8, so the results are not directly comparable. It would be good to demonstrate that the re-implementation of the baselines matches the original results.
- If space is a concern, reporting only one of mIoU or mBO in the main text would a reasonable compromise to include more datasets and baselines.

A minor issue regarding clarity: in tables 1-4, results are reported with a confidence interval. For many results, the confidence intervals of MetaSlot and the baseline overlap. In those cases, I do not think it's appropriate to mark the MetaSlot numbers in bold, as if to suggest a significant improvement.

In terms of originality, the proposed method combines existing techniques, such as slot attention and vector quantization/vocabulary learning. The experimental results are positive, but the method does not introduce a fundamentally new approach, nor it provides new theoretical insights on the learning mechanisms of object-centric methods. If the main contribution is the ability to automatically adapt the number of slots for each image, a more thorough evaluation of this aspect is needed. For example, comparing with AdaSlot in terms of performance, number of "discovered" objects, and characterization of the learned prototypes. An insightful research question could be: "for years, we have used 7 slots for COCO, 11 for CLEVR, and 6 for VOC, but how optimal is this choice?".

---

> ### Author Rebuttal · Authors · 2025-07-31
>
> We appreciate the reviewer’s constructive feedback and address the concerns as follows.
>
> >Missing comparison with AdaSlot, which is mentioned in the related work section as the most prominent method to obtain an adaptive number of slots.
>
> Thank you for the insightful suggestion regarding the comparison with AdaSlot. We did not include it in our original experimental design, as AdaSlot does not exhibit clear advantages over fixed-slot Slot Attention on real-world datasets, and in some cases even shows degraded performance. We have included a comparative experiment between MetaSlot and AdaSlot. The results align well with our findings and further support the claims made in the paper. Moreover, we report the results of the MetaSlot model using ViT-B/16 under the same training settings as in the AdaSlot paper. Due to limitations in time and computational resources, MetaSlot was trained for only 40,000 steps (compared to 500,000 steps in AdaSlot), yet it already surpasses the best performance reported in the AdaSlot paper.
>
> We speculate that this is partly because, when the number of slots varies, the model can no longer apply separate Gaussian initialization to each slot individually. In contrast, MetaSlot adopts a two-stage aggregation strategy that ensures the slot configuration in the second stage remains aligned with the Slot Attention output from the first stage. Therefore, we argue that **enabling variable slot numbers without sacrificing performance** is a more impactful contribution. We will incorporate the above discussion into the revised version of the paper.
>
> | DINOv2 ViT-S/14 on MS COCO | ARI  | FG-ARI | MBO  | mIoU |
> |----------------|------|--------|------|------|
> | AdaSlot | 17.8 | 35.1   | 27.9 | 26.8 |
> | MetaSlot | 22.4 | 40.3   | 29.5 | 27.9 |
>
> | DINOv2 ViT-B/16 on MS COCO | FG-ARI | MBO  |
> |----------------|--------|------|
> | AdaSlot (train step: 500000)| 39.0   | 27.36 |
> | BO-QSA (train step: 40000)| 37.64   | 27.07 |
> | MetaSlot (train step: 40000)| 40.13   | 27.45 |
>
> >While I agree that datasets such as CLEVR and ShapeStacks can be left out because of their simplicity, I think the MOVI datasets should be part of the evaluation.
>
> We have added experimental results where MetaSlot serves as the grouper in the SlotContrast [1] evaluation framework, evaluated on the MOVi-C dataset. In these experiments, MetaSlot does not utilize SlotContrast losses or any temporal-specific enhancements. Due to limitations in time and computational resources, we used a batch size of 8 (compared to 64 in SlotContrast), 24 frames per sample, and trained for a maximum of 20,000 steps (compared to 100,000 in SlotContrast). MetaSlot demonstrates significant performance advantages. We will include the full comparison experiments in the appendix of the revised version. The parameters will be aligned with those used in SlotContrast to ensure the experiments can be thoroughly validated by the community. We will incorporate the above discussion into the revised version of the paper.
>
> | MOVi-C         | FG-ARI | MBO  |
> |----------------|--------|------|
> | SlotContrast | 62.4 | 30.6   |
> | MetaSlot       | 63.9   | 35.0 |
>
>
> >Only DINOv2 ViT-S is used as the backbone, leaving out larger backbones and alternative models such as CLIP or MAE.
>
> Due to limited computational resources and time constraints, we were unable to include CLIP-based experiments — we apologize for this. As demonstrated in the DINOv2 paper, MAE performs worse than DINOv2 and was therefore not considered. As noted earlier, we conducted experiments using MetaSlot with DINOv2 ViT-B/16 as the encoder and compared it against AdaSlot. Notably, DINOv2 ViT-B/16 has 85.8M parameters and is approximately four times larger than ViT-S/14 in network scale. We also included additional results using ViT-S/16 to further demonstrate the generalizability of MetaSlot. Additional results with more encoders will be included in a future revision of the paper.
>
> | DINOv2 ViT-B/16 on MS COCO | FG-ARI | MBO  |
> |----------------|--------|------|
> | AdaSlot (train step: 500000)| 39.00   | 27.36 |
> | BO-QSA (train step: 40000)| 37.64   | 27.07 |
> | MetaSlot (train step: 40000)| 40.13   | 27.45 |
>
> | DINOv2 ViT-S/16 on VOC   | ARI  | FG-ARI | MBO  | mIoU |
> |----------------|------|--------|------|------|
> | BO-QSA | 20.2 | 34.5   | 39.0 | 38.0 |
> | MetaSlot | 21.7 | 36.1   | 40.2 | 39.1 |
>
> >An insightful research question could be: "for years, we have used 7 slots for COCO, 11 for CLEVR, and 6 for VOC, but how optimal is this choice?".
>
> Thank you for your suggestion. We will further explore this issue in future work. Since the codebook and the aggregator module are jointly optimized, the performance of the aggregator—such as using a slot number that significantly exceeds the maximum number of objects in a scene—can affect the convergence of the codebook, especially during the early stages of training. To address this, we have conducted an additional analysis on MetaSlot's sensitivity to different slot numbers. The results show that the optimal slot number aligns well with the long-standing empirical choices commonly adopted in the object-centric learning community.
>
> | DINOSAUR on MS COCO   | ARI  | FG-ARI | MBO  | mIoU |
> |----------------|------|--------|------|------|
> | MetaSlot, slot_num=5 | 25.5 | 37.8   | 29.3 | 27.4 |
> | MetaSlot, slot_num=7 | 22.4 | 40.3   | 29.5 | 27.9 |
> | MetaSlot, slot_num=11| 19.8 | 38.7   | 28.4 | 26.9 |
> | MetaSlot, slot_num=15| 15.9 | 36.0   | 27.2 | 26.0 |
>
> >In figure 2, the predicted masks tend to be bigger than the enclosed objects, for example around the people and the cows.
>
> We can observe that when using MetaSlot for segmentation, the upper back of the cow exhibits tighter boundaries compared to those produced by BO-QSA in figure 2. Since DINOSAUR performs reconstruction on DINO's pre-trained feature maps, the attention maps are upsampled back to the original pixel space, and the slightly oversized boundaries relative to the true pixel edges are a result of this training approach [1].
>
> In unsupervised object discovery tasks, object categories are not considered [1]. The high scores are maintained because the evaluation metrics for object discovery ignore class labels and instead use the best-matched IoU (40.3% in MetaSlot), making them more tolerant of mismatches compared to COCO mAP (64.6% in YOLOv4).
>
> >Can we simply see object-centric methods as a form of clustering? And can this perspective help us simplify existing methods rather than adding complexity?
>
> We appreciate the reviewer’s insightful observation that Object-Centric Learning (OCL) can be interpreted as a clustering process. It leverages the Slot Attention mechanism to assign image feature maps to a limited number of slots via competitive binding—these slots effectively serve as cluster centroids, akin to those in Gaussian Mixture Models. However, achieving truly object-level clustering remains a core challenge in the field, as opposed to mere aggregation of low-level features into attribute-level or category-level representations. Slot Attention introduces an inductive bias by constraining the slot dimensionality to be significantly lower than that of the original features, thereby encouraging object-level abstraction.
>
> Nonetheless, the standard formulation—using a fixed number of slots initialized from an identical isotropic Gaussian distribution—introduces two fundamental limitations:
>
> 1. **Lack of semantic prior**: Since all slots are initialized from the same isotropic Gaussian, they lack sufficient inter-cluster separation at the early stage of training. This often leads to mixed assignment where features of different objects are grouped into the same slot, or a single object is incorrectly split across multiple slots, resulting in over-/under-segmentation.
>
> 2. **Cluster-number misspecification**: A fixed slot number $K$ fails to match the varying number of true objects $N$ in dynamic scenes, leading to a model-order mismatch. When $K < N$, under-clustering (under-segmentation) occurs; when $K > N$, over-clustering (over-segmentation) arises, both of which impair object consistency.
>
> BO-QSA addresses the first issue by introducing learnable, slot-specific Gaussian priors—effectively planting $K$ more dispersed centroids during initialization, which alleviates semantic entanglement and helps stabilize background slots. However, the diversity of these priors remains limited by the fixed number of slots, which still constrains their effectiveness in real-world scenarios with high object diversity and variable object counts (e.g., COCO, VOC).
>
> MetaSlot overcomes both longstanding challenges in OCL by introducing a two-stage aggregation mechanism guided by prototype slots. Rich semantic priors are injected early on via these prototype slots, providing high-resolution object-level cues and enabling the initial centroids to exhibit clearer inter-class separation. At the same time, the prototype matching mechanism addresses cluster-number misspecification by allowing flexible alignment between dynamic object counts and slot representation.
>
> Furthermore, we inherit the bi-level optimization strategy from BO-QSA: the two-stage Slot Attention executes its full iterative process, but gradients are computed and backpropagated only at the final update step of the second-stage MSA. As a result, parameter updates rely solely on stabilized slot assignments, thereby reducing gradient noise induced by cluster-number mismatch and improving convergence efficiency.
>
> > Minor points
>
> Finally, thank you for your valuable suggestions regarding the details of our paper. Additionally, in MetaSlot, the Slot Attention (SA) used in the first stage and the Masked Slot Attention (MSA) used in the second stage share the same parameters. Therefore, MSA serves as the key component in enabling support for a dynamic number of slots.
>
> [1] Seitzer, Maximilian, et al. Bridging the Gap to Real-World Object-Centric Learning, ICLR 2023

---

> > ### Comment · Reviewer_ayFM · 2025-08-08
> > **Discussion**
> >
> > Dear authors, thanks for the well-written rebuttal.
> >
> > My initial review was already positive and the rebuttal addressed my concerns/questions. No further discussion needed from my side, I will update my rating.
> >
> > Please don't forget to integrate all the additional discussion and experimental details in the paper.

---

> > > ### Author Response · Authors · 2025-08-08
> > >
> > > Thank you for your positive feedback and for acknowledging our efforts. We are glad to hear that your concerns have been addressed. We will make sure to integrate all the additional discussion and experimental details into the final revised version of the paper.

---

### Official Review · Reviewer_Y71X · 2025-06-30

**Clarity:** 2
**Significance:** 3
**Originality:** 2
**Rating:** 4
**Confidence:** 4

**Summary:**

The paper proposes MetaSlot, a Slot Attention variant for object-centric learning that enables dynamic slot allocation through a vector-quantized (VQ) codebook of object prototypes. The method introduces a two-stage aggregation process: an initial stage aligns slots with prototypes and prunes duplicates, while a second refinement stage injects progressively annealed noise and uses masked attention.

**Questions:**

"Optimal" slot number: Can the authors provide a more principled discussion of what constitutes an optimal number of slots? Or at least clarify how MetaSlot approximates this adaptively? A more thorough discussion of codebook dynamics, computational trade-offs, and theoretical understanding of slot allocation would strengthen the paper.

**Ethical Concerns:**

["NO or VERY MINOR ethics concerns only"]

**Final Justification:**

The rebuttal satisfactorily addresses certain concerns and partially mitigates the issues I raised. Nonetheless, important questions remain regarding the work’s originality, the absence of rigorous theoretical analysis, and ambiguities in the reported computational costs. Considering the partial resolution of my concerns, as well as the generally positive assessments from other reviewers, I will modestly increase my score.

**Limitations:**

yes

**Paper Formatting Concerns:**

No major formatting issue.

**Quality:**

2

**Strengths And Weaknesses:**

1. Strengths:
- Tackles a real limitation of traditional Slot Attention by removing the constraint on fixed slot numbers.
- Uses a plug-and-play design that can be integrated with multiple object-centric frameworks (Transformer, MLP, diffusion).
- Empirical results show consistent improvements in segmentation and representation quality.

2. Weaknesses:
- Limited originality: The method is largely a recombination of known techniques (VQ, masking, noise annealing) applied to Slot Attention. The dynamic slot count is driven by prototype quantization, which is conceptually straightforward.
- Overstated claims: Several statements (e.g., line 24–25) present strong assertions about the necessity of discrete object-level representations without theoretical or empirical support. Recent large vision–language models (e.g., GPT-4) achieve strong zero-shot reasoning without an explicit object slot bottleneck. Their success shows that such reasoning can also emerge from dense, distributed features, so the claim should be softened or empirically justified.
- No theoretical analysis: The paper lacks discussion or formal insight into when dynamic slot counts are beneficial, or what constitutes an "optimal" slot number. Over-segmentation is assumed to be harmful without deeper justification.
- No discussion of computational cost: The two-stage aggregation and codebook updates introduce overhead, but FLOPs or throughput are not reported. Performance gains may be partially attributable to increased compute.
- Codebook sensitivity unexplored: The size of the prototype codebook (fixed to 512) is not studied, despite being critical to slot granularity. It would be helpful to conduct an ablation on the number of prototypes and provide insight into how it affects the number and granularity of resulting slots—e.g., do more prototypes lead to finer, potentially redundant slots, while fewer prototypes yield coarser but more stable object groupings? This is important for understanding and controlling the behavior of MetaSlot.
- Writing clarity issues: Some equations (e.g., Eq. (1)) are needlessly complicated, and some passages are vague or imprecise.

---

> ### Author Rebuttal · Authors · 2025-07-31
>
> We appreciate the reviewer’s constructive feedback and address the concerns as follows.
>
> >Limited originality
>
> To the best of our knowledge, we are the first to leverage vector quantization (VQ) to learn more expressive query slots in slot-based models, whereas existing approaches typically rely on sample-independent Gaussian priors. We are also the first to introduce a noise annealing strategy to improve the iterative dynamics of Slot Attention. Furthermore, we are the third to implement object-centric learning with a variable number of slots (following SOLV and AdaSlot), while there are nearly a thousand papers based on Slot Attention that still rely on a fixed number of slots. Among them, SOLV adopts a post-hoc clustering method rather than end-to-end learning and requires careful hyperparameter tuning. AdaSlot, despite introducing dynamic slot counts, exhibits a slight performance drop compared to its fixed-slot counterpart. In contrast, we provide comparative experiments showing that MetaSlot achieves substantial performance gains over AdaSlot.
>
> Therefore, we argue that achieving substantial improvements while supporting **a variable number of slots—without compromising performance**—represents a more impactful contribution to the field.
>
> Moreover, in slot-based object-centric learning (OCL), introducing a quantization module to enhance the Slot Attention mechanism is neither straightforward nor intuitive. This is primarily because such methods are inherently unsupervised and lack any form of labeled data, often resulting in suboptimal or insufficient slot representations. Additionally, the semantic information captured by Slot Attention during early training stages is usually ambiguous. Using these ambiguous representations to update the codebook, which in turn influences Slot Attention, can easily lead to mode collapse. Therfore, MetaSlot’s ability to successfully incorporate vector quantization, provide more effective prior guidance, and support dynamic slot adaptation is the result of deliberate and careful design. This includes key components such as Mask Slot Attention, Noise Injection, and the Two-step Aggregation mechanism.
>
> It is also worth noting that the vector quantization strategy adopted in MetaSlot is fundamentally different from those used in existing OCL models. Due to space limitations, we respectfully refer the reviewer to our response to Reviewer gev3 for a more detailed discussion on vector quantization. We will also include a comprehensive elaboration on object-centricity, vector quantization, and slot-based OCL in the appendix of the revised version.
>
> | DINOv2 ViT-S/14 on MS COCO | ARI  | FG-ARI | MBO  | mIoU |
> |----------------|------|--------|------|------|
> | AdaSlot | 17.8 | 35.1   | 27.9 | 26.8 |
> | MetaSlot | 22.4 | 40.3   | 29.5 | 27.9 |
>
> >Overstated claims
>
> Thank you for your valuable feedback. We will revise our original discussion and incorporate additional relevant academic works to further support our claims and ensure the rigor of our arguments.
>
> Firstly, a core feature of high-level cognition is widely considered to be the ability to systematically manipulate and compose discrete units of knowledge [1]. While structured representations naturally emerge as tokens in text, obtaining analogous discrete representations in unstructured modalities like images remains a fundamental challenge [2]. This limits current generative models in understanding fine-grained object interactions and modeling physical relationships. As such, Object-Centric Learning (OCL) primarily focuses on the upstream challenge of representation construction, rather than directly addressing downstream tasks involving complex reasoning.
>
> Secondly, recent advances in multimodal foundation models have demonstrated that object-centric representations can significantly enhance model understanding [3, 4].  Moreover, a substantial body of empirical research in robotic manipulation has shown that adopting discrete, object-oriented visual representations greatly enhances a robot’s ability to perceive, plan, and act in cluttered and dynamic environments[5].
>
> >No theoretical analysis
>
> This direction currently lacks a well-established theoretical framework, making theoretical analysis challenging; we plan to explore this in future work. However, regarding the impact of slot count on performance, we can provide a more in-depth experimental analysis as follows. Since the codebook and the aggregator module are jointly optimized, the performance of the aggregator can influence the convergence of the codebook, especially during the early stages of training. Therefore, we  found that the optimal slot count aligns with the long-standing empirical choices commonly adopted by the community.
>
> | DINOSAUR on MS COCO   | ARI  | FG-ARI | MBO  | mIoU |
> |----------------|------|--------|------|------|
> | MetaSlot, slot_num=5 | 25.5 | 37.8   | 29.3 | 27.4 |
> | MetaSlot, slot_num=7 | 22.4 | 40.3   | 29.5 | 27.9 |
> | MetaSlot, slot_num=11| 19.8 | 38.7   | 28.4 | 26.9 |
> | MetaSlot, slot_num=15| 15.9 | 36.0   | 27.2 | 26.0 |
>
> >No discussion of computational cost
>
> Regarding the concern about computational cost, we report a detailed comparison between MetaSlot and the original Slot Attention in the table below:
>
> | Model          | Parameters (M) | FLOPs (G) | GPU Memory (MB) |
> | -------------- | -------------- | --------- | --------------
> | MetaSlot       | 33.1           | 44.2      |558
> | Slot Attention | 33.0           | 43.9      |554
>
> The difference between the two is minimal (less than 1%), indicating that the significant performance gains of MetaSlot are not due to increased computational resources, but rather stem from our proposed structural innovations—such as Mask Slot Attention and the two-step aggregation mechanism.
>
> In addition, we note that previous literature in object-centric learning has consistently shown that merely increasing the number of Slot Attention iterations does not yield meaningful performance improvements. To further verify this, we conducted an additional ablation study on the BO-QSA model, increasing its iteration count to 6 to match that of MetaSlot. The results clearly show that increasing the iteration count alone did not improve BO-QSA’s performance.
>
> | DINOSAUR on MS COCO   | ARI  | FG-ARI | MBO  | mIoU |
> |----------------|------|--------|------|------|
> | MetaSlot | 22.4 | 40.3   | 29.5 | 27.9 |
> | BO-QSA, iter_num=6 | 17.1 | 37.9 | 27.7 | 26.3 |
>
> >Codebook sensitivity unexplored
>
> We conducted an in-depth exploration and analysis of different codebook sizes, including 256, 512, and 1024. Based on experimental results and visualizations, we observed that a smaller codebook size (e.g., 256) tends to cause slight under-segmentation, as reflected by a marginal improvement in ARI but a slight drop in FG-ARI. When increasing the size to 1024, we found that ARI slightly decreased while FG-ARI improved compared to the 512-sized version. This is because **the number of prototype slots (256, 512, 1024) is significantly smaller than the total number of objects in the dataset** (about 2.5 million). As a result, **increasing the number of prototypes does not lead to severe over-segmentation, and all variants in this ablation study still outperform the original Slot Attention by a clear margin.**
>
> Moreover, we observed that a larger codebook (e.g., size 1024) allows the model to distinguish between spatially distinct but semantically similar slots more precisely. For example, one prototype may specialize in representing horses in the upper-left region of an image, while another captures horses in the lower-left region. A larger codebook also enables finer-grained object categorization, such as further differentiating dials into circular and square types. However, it may also introduce minor over-segmentation effects, such as isolating the head of a person into a separate slot. That said, this level of over-segmentation is generally acceptable in practical applications, as the model still maintains strong semantic consistency across finer divisions—unlike forced over-segmentation caused by severe mismatch between object and slot numbers.
>
> This phenomenon also reflects a broader challenge in the object-centric learning community: the difficulty of clearly defining what constitutes an "object." However, this issue lies beyond the scope of our current MetaSlot paper.
>
> | DINOSAUR on MS COCO         | ARI  | FG-ARI | MBO  | mIoU |
> |----------------|------|--------|------|------|
> | MetaSlot, codebook_size=256 | 26.7 | 38.1   | 29.4 | 27.5 |
> | MetaSlot, codebook_size=512 | 22.4 | 40.3   | 29.5 | 27.9 |
> | MetaSlot, codebook_size=1024| 20.0 | 40.4   | 29.2 | 27.3 |
>
> >Writing clarity issues
>
> Thank you for raising the concern regarding writing clarity. Most of the equations in our paper follow conventions commonly adopted in the object-centric learning community—for instance, Equation (1) is consistent with the Slot Attention formulation used in BO-QSA. That said, we sincerely welcome any suggestions for improvement. If you could kindly specify which equations or sections felt unclear, we would be more than happy to revise them to improve clarity and readability.
>
> [1] Lake, Brenden M., et al. Building machines that learn and think like people, Behavioral and brain sciences 40 2017
>
> [2] Greff, Klaus, Sjoerd Van Steenkiste, and Jürgen Schmidhuber. On the Binding Problem in Artificial Neural Networks, preprint 2020
>
> [3] Wang, Shihao, et al. Omnidrive: A holistic llm-agent framework for autonomous driving with 3d perception, reasoning and planning. CoRR 2024.
>
> [4] Zhang, Tao, et al. Omg-llava: Bridging image-level, object-level, pixel-level reasoning and understanding, NeurIPS 2024
>
> [5] Shi, Junyao, et al. Composing Pre-Trained Object-Centric Representations for Robotics From “What” and “Where” Foundation Models, ICRA 2024

---

> ### Comment · Reviewer_Y71X · 2025-08-05
>
> The rebuttal addresses several points, including ablations on codebook size and a response to computational cost. However, core concerns remain unresolved. The paper still lacks theoretical grounding or principled analysis of key design choices, such as how the number of slots should be selected or how the quantization dynamics influence model behavior. In terms of novelty, the method primarily combines existing components, and the conceptual contribution remains limited.
>
> Moreover, the response on computational efficiency raises further concerns. The reported GPU memory usage (~550MB) is implausibly low for a model with 33M parameters and 44G FLOPs. Such memory usage does not match expected values under realistic training settings, and likely omits activation memory or runtime overhead. The absence of details such as batch size, input resolution, or profiling methodology further reduces the credibility of this analysis. These limitations reduce the clarity, depth, and significance of the contribution.

---

> > ### Author Response · Authors · 2025-08-05
> >
> > We appreciate the reviewer's thoughtful comments and valuable insights. We believe our rebuttal has effectively clarified the novel contributions of MetaSlot. Specifically, regarding the number of objects, we emphasize that MetaSlot's joint optimization sets an upper bound for slot count that aligns naturally with widely accepted practices in the community.  Additionally, we have thoroughly examined how varying codebook sizes impact performance, offering detailed empirical evidence alongside theoretical reasoning to substantiate our claims.
> >
> > Concerning GPU memory usage, we wish to clarify that the reported 550MB figure specifically represents inference memory usage for a single 224×224 image, which is inherently independent of batch size. During training, we employed a batch size of 32 along with the DINOv2 ViT-S/14 backbone, resulting in a GPU memory consumption of approximately 16GB. We apologize if our original explanation was unclear and hope this addresses the reviewer's concerns regarding the reported figures.
> >
> > Finally, we assert that our comparison between MetaSlot and the baseline model using an equal number of iterations (iter=6), and therefore consistent computational complexity, robustly demonstrates MetaSlot’s performance improvements. Furthermore, it is a well-established notion within the object-centric learning literature that merely increasing iteration count—and thus computational complexity—does not necessarily translate to improved performance, a point supported by numerous prior ablation studies.
> >
> > We sincerely thank the reviewer again for the constructive feedback, which has enabled us to strengthen the clarity and impact of our work.

---

### Official Review · Reviewer_MgVn · 2025-06-30

**Clarity:** 3
**Significance:** 3
**Originality:** 3
**Rating:** 5
**Confidence:** 5

**Summary:**

The paper proposes to use a set of prototypes for making SA-based OCL models (including real-world ones such as DINOSAUR, VideoSAUR, and SPOT), more flexible in terms of the number of slots and not restricted by random initialization that puts all the burden on iterative SA decomposition of the scene. It shows nice improvements for Video-based OCL that also requires slots to be consistent in time. To achieve this, the paper employs a two-step procedure: applying the same slot attention module to random slots and then to masked discrete prototypes, which are obtained as a batch k-means over slots. To align the second state of SA, the paper is adding decreasing noise to the inputs.

**Questions:**

I believe that the proposed approach is a simple yet meaningful way to incorporate flexibility in terms of the number of slots into OCL.  My primary concern is that the presentation and experimental analysis could be improved, making it more helpful to the community. If the paper can address most of the weaknesses mentioned, I would be happy to increase my score.

**Ethical Concerns:**

["NO or VERY MINOR ethics concerns only"]

**Final Justification:**

After reading the answers from the rebuttal, I think the authors addressed my main concerns and I'm happy to increase my score towards acceptance.

"Due to limitations in time and computational resources..."
If accepted, I encourage authors to run full experiments for camera ready, not a shorter one, to make sure that the comparison with other methods are valid.

**Limitations:**

yes

**Quality:**

3

**Strengths And Weaknesses:**

### Strengths:

1. The paper is generally well-written and well-structured.
2. The paper evaluates the quality of the slots on different downstream tasks, not only scene decomposition, but also the content of the features and not only the quality of the slots.
3. Visualization of the meta slots (Figure 3) is nice for understanding the reasons for the method's performance improvements.

### Weaknesses:
1. Some presentations still can be improved. For example, if I understand correctly, the Masked Slot Attention (without noise) is basically equivalent to running fewer slots as inputs, while a masking is needed to make this procedure efficient for batch processing. Is this a case? If yes, the presentation could be improved accordingly. If no, it would be great to discuss or note what the difference is.  Also the name is a bit misleading given the current analysis: from the name it seems like paper tried to solve only one issue of the number of slots, while during reading it becomes clear that it is tacking both interesting and flexible semantic initialization and also number of slots issue.

Additionally, there are some related works that are missing, such as SlotContrast [1], which observed automatic adaptation to the number of slots due to a contrastive objective and achieved a state-of-the-art result in video object-centric learning on YT-VIS.

2. Although some ablations are performed, they are still insufficient to fully understand the performance improvements.  So I suggest some additional ablations: what if we use random init in the second iteration and just change the number of slots (e.g use masked SA with masks for slots that were mapped to the same prototype). Another ablation could be if we are not removing outdated prototypes (Eq 11.)
3. While the main point of MetaSlot is to improve any OCL method, it is still important to know if it is possible to improve state-of-the-art OCL learning. For example, it would be beneficial to compare MetaSlot with SlotContrast [1] to determine if MetaSlot further enhances the contrastive slots and masks of SlotContrast. Especially, it is important that the comparison with SPOT is a bit mixed. While authors showed that MetaSlot+one Stage SPOT is comparable with two-stage SPOT, they didn't show that they can improve upon two-stage SLOT by adding MetaSlot.
4. Number of slots analysis: Direct analysis of the number of slots per image vs GT number of objects is missing (similar to AdaSlot). Although their analysis is somewhat limited due to the number of background slots, it remains quite informative. Additionally, the paper would benefit from a sensitivity analysis (e.g., by varying the maximum number of slots used or the number of prototypes used). Would we observe that more slots are turned off, and that the results are still similar, or would we still oversegment?

[1] Manasyan A, et al. Temporally Consistent Object-Centric Learning by Contrasting Slots, CVPR 2025

---

> ### Author Rebuttal · Authors · 2025-07-31
>
> We appreciate the reviewer’s constructive feedback and address the concerns as follows.
>
> >The Masked Slot Attention (without noise) is basically equivalent to running fewer slots as inputs, while a masking is needed to make this procedure efficient for batch processing. Is this a case?
>
> Yes, your understanding is absolutely correct. In MetaSlot, the Slot Attention (SA) used in the first stage and the Masked Slot Attention (MSA) employed in the second stage share exactly the same parameters. Therefore, they are two consecutive calls to the same SA module rather than two independent aggregators. Consequently, we keep the same upper bound on the number of slots in the MSA stage and rely on binary masks to deactivate redundant slots, thereby enabling variable slot counts while preserving batch‑level efficiency on the GPU.
>
> >From the name it seems like paper tried to solve only one issue of the number of slots, while during reading it becomes clear that it is tacking both interesting and flexible semantic initialization and also number of slots issue.
>
> Thank you for recognizing the MetaSlot mechanism! Regarding the choice of our paper's title, we emphasized “variable number of slots” because a fixed number of slots has long been considered a core bottleneck in slot-based unsupervised learning, with very limited existing solutions. To date, the only method that truly attempts dynamic slot allocation is AdaSlot. However, its performance on real-world datasets (such as COCO and VOC) in terms of MBO, ARI, and FG ARI metrics not only fails to surpass that of fixed-slot Slot Attention (SA), but in fact, performs worse.
>
> We speculate that this is partly because, when the number of slots varies, the model can no longer apply the Gaussian initialization provided by BO-QSA to each slot individually. As a result, the initial binding between semantic and positional information is weakened, negatively affecting performance. Additionally, AdaSlot predicts object count directly from the feature map, which makes it highly sensitive to the weight of the regularization loss.
>
> In contrast, MetaSlot employs a two-stage aggregation strategy. Its mask indices are derived from codebook matching, ensuring that the slot layout in the second stage remains consistent with the Slot Attention output from the first stage. This prevents confusion in slot-object binding. Given this landscape, we argue that achieving substantial improvements while **supporting a variable number of slots—without compromising performance**—represents a more impactful contribution to the field.
>
> >There are some related works that are missing, such as SlotContrast.
>
> We conducted video object-centric learning experiments on YT-VIS primarily to demonstrate the applicability of MetaSlot in video settings. It is important to note that the current version of MetaSlot has not been specifically designed with additional temporal modeling for video scenarios.
> We have added experimental results where MetaSlot serves as the grouper in the SlotContrast evaluation framework, evaluated on the MOVi-C dataset. In these experiments, MetaSlot does not utilize SlotContrast losses or any temporal-specific enhancements. Due to limitations in time and computational resources, we used a batch size of 8 (compared to 64 in SlotContrast), 24 frames per sample, and trained for a maximum of 20,000 steps (compared to 100,000 in SlotContrast). MetaSlot still demonstrates significant performance advantages. We will include the full comparison experiments in the appendix of the revised version. The parameters will be aligned with those used in SlotContrast to ensure the experiments can be thoroughly validated by the community.
>
> | MOVi-C         | FG-ARI | MBO  |
> |----------------|--------|------|
> | SlotContrast | 62.4 | 30.6   |
> | MetaSlot       | 63.9   | 35.0 |
>
> >Although some ablations are performed, they are still insufficient to fully understand the performance improvements.
>
> In our previous ablation studies, we examined the impact of MetaSlot’s masking mechanism and noise injection mechanism, demonstrating that appropriate slot assignment can significantly enhance model performance. Moreover, even when the masking mechanism is removed, MetaSlot still outperforms the original Slot Attention, which we believe provides indirect evidence of the effectiveness of the prototype-guided mechanism.
>
> We also added two new experiments to gain deeper insight into the working mechanism of MetaSlot:
> (1) Removing the prototype guidance and instead initializing slots in the second stage by sampling from separate Gaussian distributions, as done in the first stage;
> (2) Disabling the reactivation mechanism for stale(dead) prototypes during the codebook update process.
>
> These results further support the effectiveness of our module design and align with the theoretical assumptions proposed in the paper.
>
> | Setting         | ARI  | FG-ARI | MBO  | mIoU |
> |----------------|------|--------|------|------|
> | MetaSlot (full) | 22.4 | 40.3   | 29.5 | 27.9 |
> | w/o proto       | 21.1 | 40.2   | 29.1 | 27.6 |
> | w/o prune       | 21.7 | 40.0   | 29.1 | 27.5 |
>
> >While the main point of MetaSlot is to improve any OCL method, it is still important to know if it is possible to improve state-of-the-art OCL learning.
>
> To objectively compare the performance of different methods, we adopted a unified, custom-built evaluation framework (We will release the evaluation framework as open source in the future.). It is worth noting that Spot's two-stage distillation strategy is highly sensitive to hyperparameters and often requires additional tuning, which is beyond the scope of what MetaSlot aims to address. For example, SPOT applies gradient clipping using the infinity norm with a maximum value of 0.3, whereas most mainstream OCL methods use the L2 norm with a maximum value of 1. In our experiments integrating various encoder-decoder architectures—including VQ-VAE, DVAE, diffusion models, and Transformers—MetaSlot consistently achieved significant and stable performance improvements, further validating its effectiveness and broad applicability.
>
> Additionally, SlotContrast’s contrastive loss was originally designed to work with a fixed number of slots, and extending it to support variable slot counts would require additional effort. However, as previously discussed, even without incorporating SlotContrast’s core contribution—the contrastive loss—MetaSlot demonstrates comparable or even superior performance in object discovery tasks. We believe that combining SlotContrast’s contrastive learning objective in the first stage with MetaSlot’s dynamic aggregation in the second stage could yield further improvements, particularly for temporal object discovery tasks. Nonetheless, this would entail architectural modifications that fall outside the scope of the current work and will be explored in future research.
>
> In summary, our core contribution is the proposal of MetaSlot, a general-purpose variant of Slot Attention. Further optimizations for temporal modeling can be explored in future work, following the evolution path of VideoSaur from DINOSAUR or SAVi from the original Slot Attention.
>
> >Direct analysis of the number of slots per image vs GT number of objects is missing. The paper would benefit from a sensitivity analysis.
>
> We analyzed the number of objects discovered by MetaSlot on the COCO dataset. Given the maximum slot count was capped at 7, the resulting distribution exhibited a skewed shape with a mean of 4.9, a standard deviation of 1.2, and a range from 1 to 7. In contrast, the ground-truth object count in COCO follows a long-tailed distribution—with a mean of 6.3 (elevated due to the long-tail nature of the data), a standard deviation of 6.4, and a range from 0 to 63. Visualization results show that MetaSlot approximates the real object count distribution curve reasonably well, despite the capped slot limit. Additionally, we have included comparative experiments with AdaSlot, and the results remain consistent with the statements in the main paper.
>
> Furthermore, we investigated the impact of different codebook sizes (256, 512, and 1024) on model performance. We observed that codebook size has a limited effect on performance; however, a larger prototype set allows for finer distinctions between slots that share semantic concepts but differ in spatial location. For example, as shown in Figure 3, slot #445 corresponds to the concept of “person” located on the right side of the image, while slot #337 represents the same concept but appears at the center of the scene.
>
> We also examined how the number of slots affects final performance. Since the codebook and the aggregator module are jointly optimized, the quality of the aggregator—particularly in early training stages—can significantly influence codebook convergence. We found that the empirically optimal slot count aligns well with long-standing choices commonly adopted in the community.
>
> All of the above analyses and results will be included in the appendix of the revised paper, where we provide a more detailed discussion on MetaSlot’s capability in estimating object counts, along with extended comparative experiments and supporting analyses.
>
> | DINOSAUR on MS COCO         | ARI  | FG-ARI | MBO  | mIoU |
> |----------------|------|--------|------|------|
> | MetaSlot, codebook_size=256 | 26.7 | 38.1   | 29.4 | 27.5 |
> | MetaSlot, codebook_size=512 | 22.4 | 40.3   | 29.5 | 27.9 |
> | MetaSlot, codebook_size=1024| 20.0 | 40.4   | 29.2 | 27.3 |
> | MetaSlot, slot_num=5 | 25.5 | 37.8   | 29.3 | 27.4 |
> | MetaSlot, slot_num=7 | 22.4 | 40.3   | 29.5 | 27.9 |
> | MetaSlot, slot_num=11| 19.8 | 38.7   | 28.4 | 26.9 |
> | MetaSlot, slot_num=15| 15.9 | 36.0   | 27.2 | 26.0 |
>
> | DINOSAUR on MS COCO   | ARI  | FG-ARI | MBO  | mIoU |
> |----------------|------|--------|------|------|
> | AdaSlot | 17.8 | 35.1   | 27.9 | 26.8 |
> | MetaSlot | 22.4 | 40.3   | 29.5 | 27.9 |

---

> > ### Comment · Reviewer_MgVn · 2025-08-06
> > **Rebuttal by Authors responce**
> >
> > I thank the authors for their time addressing my concerns. In particular, it is great to see additional analysis for the number of slots and codebook size and comparison with AdaSlot. Also, I think it is useful for readers to see MetaSlot in Video Regime (i.e, comparison with SlotContrast).
> >
> >
> > After reading the answers from the rebuttal, I think the authors addressed my main concerns and I'm happy to increase my score towards acceptance.

---

> > > ### Author Response · Authors · 2025-08-06
> > >
> > > Thank you very much for your positive feedback and valuable suggestions. We are pleased to hear that your main concerns have been addressed. We will further enrich the comparison in the video regime in our revised manuscript. Thank you again for your constructive comments and support.

---

### Decision · Program_Chairs · 2025-09-17

**Decision:**

Accept (poster)

**Comment:**

This paper studies the problem of determining the number of components (or slots) in object-centric learning. The method, which is based on a VQ codebook of object prototypes, can be applied to several existing frameworks to yield improvements as is shown. Improvements are shown on real-world datasets, thereby contributing to the state-of-the-art in this area.

Reviewers were initially largely positive about the contribution, though there were some issues raised, such as about comparisons to other methods for object-centric learning, or further ablations and analysis, which the authors addressed during the rebuttal. One reviewer lists limited novelty and lack of theoretical analysis as a remaining concern that wasn't fully addressed, even after the rebuttal. However, no major issues remain, and all reviewers are in favor of acceptance.

In summary, the AC agrees with the reviewers that this is a valuable contribution that will certainly be of interest to object-centric learning, and that (after rebuttal) the quality is sufficient to warrant acceptance.